# Conditional blastocyst complementation of a defective Foxa2 lineage efficiently promotes the generation of the whole lung

Akihiro Miura[1,2], Hemanta Sarmah[1], Junichi Tanaka[1], Youngmin Hwang[1], Anri Sawada[1], Yuko Shimamura[1], Takehiro Otoshi[1], Yuri Kondo[1], Yinshan Fang[1], Dai Shimizu[1], Zurab Ninish[1], Jake Le Suer[3,4], Nicole C Dubois[5], Jennifer Davis[6], Shinichi Toyooka[2], Jun Wu[7], Jianwen Que[1], Finn J Hawkins[3,4], Chyuan-Sheng Lin[8], Munemasa Mori[1]*

[1]Columbia Center for Human Development and Division of Pulmonary, Allergy, Critical Care, Department of Medicine, Columbia University Medical Center, New York, United States; [2]Department of Thoracic, Breast and Endocrinological Surgery, Okayama University Graduate School of Medicine, Dentistry and Pharmaceutical Sciences, Okayama, Japan; [3]The Pulmonary Center and Department of Medicine, Boston University School of Medicine, Boston, United States; [4]Center for Regenerative Medicine, Boston University and Boston Medical Center, Boston, United States; [5]Department of Cell, Developmental and Regenerative Biology, Icahn School of Medicine at Mount Sinai, New York, United States; [6]Department of Pathology, University of Washington, Seattle, United States; [7]Department of Molecular Biology, University of Texas Southwestern Medical Center, Dallas, United States; [8]Bernard and Shirlee Brown Glaucoma Laboratory, Department of Pathology and Cell Biology, College of Physicians and Surgeons, Columbia University Irving Medical Center, New York, United States

*For correspondence:
mm4452@cumc.columbia.edu

**Competing interest:** The authors declare that no competing interests exist.

**Abstract** Millions suffer from incurable lung diseases, and the donor lung shortage hampers organ transplants. Generating the whole organ in conjunction with the thymus is a significant milestone for organ transplantation because the thymus is the central organ to educate immune cells. Using lineage-tracing mice and human pluripotent stem cell (PSC)-derived lung-directed differentiation, we revealed that gastrulating Foxa2 lineage contributed to both lung mesenchyme and epithelium formation. Interestingly, Foxa2 lineage-derived cells in the lung mesenchyme progressively increased and occupied more than half of the mesenchyme niche, including endothelial cells, during lung development. *Foxa2* promoter-driven, conditional Fgfr2 gene depletion caused the lung and thymus agenesis phenotype in mice. Wild-type donor mouse PSCs injected into their blastocysts rescued this phenotype by complementing the Fgfr2-defective niche in the lung epithelium and mesenchyme and thymic epithelium. Donor cell is shown to replace the entire lung epithelial and robust mesenchymal niche during lung development, efficiently complementing the nearly entire lung niche. Importantly, those mice survived until adulthood with normal lung function. These results suggest that our Foxa2 lineage-based model is unique for the progressive mobilization of donor cells into both epithelial and mesenchymal lung niches and thymus generation, which can provide critical insights into studying lung transplantation post-transplantation shortly.

## Editor's evaluation

This study clearly shows the ability to generate whole lungs using a blastocyst complementation assay. The work elegantly uses a Foxa2 deficient by ground to promote generation of lungs from a Foxa2 replete donor. These findings will spur further interest in xenotransplantation approaches for human organs.

## Introduction

Whole organ generation to treat various intractable diseases has long been challenging (*Hackett et al., 2010*; *Kemter et al., 2020*; *Ott et al., 2010*; *Petersen et al., 2010*; *Wang, 2019*). Organ bioengineering strategy based on recellularizing tissue-specific progenitors into the decellularized scaffolds, induced pluripotent stem cell (iPSC)-derived organoids, or 3D bioprinters are the next-generation tissue transplant therapies (*Guyette et al., 2014*; *Kotton and Morrisey, 2014*; *Petersen et al., 2010*; *Tian et al., 2021*). Even with these techniques, however, the mammalian lung is one of the most challenging organs to replicate because of its anatomical complexity and cellular diversity. It contains hundreds of airway branches and a thin micron-sized alveolar layer of inflated and well-vascularized alveoli composed of billions of cells from more than 50 different cell types (*Crapo et al., 1982*; *Kotton and Morrisey, 2014*; *Stone et al., 1992*; *Travaglini et al., 2020*). Donor organs for lung transplantation are in short supply worldwide, but the technology does not exist to generate whole lungs composed of tissue-specific epithelial and mesenchymal cells, including endothelial cells. Lungs grow fully only through natural lung development.

During development, lung epithelial and mesenchymal precursors interact to initiate an elaborate developmental program of organogenesis that includes specification, pattern formation, progenitor cell expansion, and differentiation. The lung epithelial cells are derived from the foregut, definitive endoderm (DE) derivatives, classically labeled by Sox17 and Forkhead Box A2 (Foxa2) (*Green et al., 2011*; *Huang et al., 2014*). Multiple genetic studies using Sonic Hedgehog (Shh) Cre lineage-tracing mice have also shown that the entire Nkx2-1+ lung and tracheal epithelial primordium arises from Shh+ DE (*Cardoso and Kotton, 2008*; *Christodoulou et al., 2011*; *Harris et al., 2006*; *Kadzik and Morrisey, 2012*; *Tian et al., 2011*; *Weaver et al., 1999*; *Xing et al., 2008*).

The lung mesenchyme primordium is derived from Wnt2+ Isl1+ cardiopulmonary progenitors (CPP) (*Peng et al., 2013*). CPP is the derivative of Osr1+ Nkx6-1+Barx1− Wnt4low foregut lung mesoderm that arises from lateral plate mesoderm (LPM) (*Han et al., 2020*). While DE and LPM arise from primitive streaks (PS) during gastrulation, the exact lineage origin of LPM has been a complete mystery.

Mesendoderm is a bipotent transitional state between the PS and nascent mesoderm labeled by Mixl1, Pdgfrα, and Brachyury (T) during gastrulation that can give rise to both DE and mesoderm (*Hart et al., 2002*; *Tada et al., 2005*). Although it was speculated that mesendoderm might form LPM and DE, there have been no conclusive genetic studies on whether mesendoderm gives rise to both lung epithelium and mesenchyme. Pdgfrα is expressed in the epiblast-derived mesendoderm, the primitive endoderm (PrE), and its extra-embryonic endoderm derivatives, such as parietal and visceral endoderm, around E5.5–E7.5. Foxa2 plays a pivotal role in alveolarization and airway goblet cell expansion (*Wan et al., 2004*), while there was a significant knowledge gap regarding Foxa2 lineage during lung development.

Blastocyst complementation (BC) has been proposed as a promising option for tissue-specific niche complementation (*Chen et al., 1993*). This unique technology has been further developed into intra- and interspecies organ generations such as kidney, pancreas, thymus, and blood vessels (*Hamanaka et al., 2018*; *Kobayashi et al., 2010*; *Usui et al., 2012*; *Yamaguchi et al., 2017*). However, co-generation of targeted organs and thymus has never been reported, while educating host immune cells for organ transplantation is critical to avoid graft-versus-host disease (GvHD) post-transplantation (*Bos et al., 2022*). The production of entire organs, including tissue-specific epithelium and mesenchyme, including endothelium, was also tricky because the origin of pulmonary endothelium seems similar to the other organs. Unfortunately, even with BC, the lungs produced were non-functional and very inefficient, and in addition, the chimeric lungs contained a substantial amount of host-derived tissue (*Kitahara et al., 2020*; *Wen et al., 2021*). Previously, we established the conditional blastocyst complementation (CBC) approach, which targets specific lineages complemented by donor PSCs (*Mori et al., 2019*). Using lineage-specific drivers of lung endoderm in CBC avoids the effects

of genetic manipulation in non-target organs for the generation of empty organ niches that lead to functional chimeric lung generation (*Mori et al., 2019*). However, most of the lung mesenchyme and endothelium were still derived mainly from the host cells, which was the severe limitation of CBC (*Mori et al., 2019*). Given that the CBC approach targeted endodermal lungs, we speculated that this limitation was due to a significant gap in our knowledge of the origin of all lung cell types, especially pulmonary mesenchyme, including endothelium. In particular, the complementation of endothelium is a critical issue for overcoming hyperacute rejection after lung transplantation. To overcome this critical issue, we explored the origin and the program of whole lung epithelium and mesenchyme, the major components of the lung.

We hypothesized that targeting a single lung precursor lineage may facilitate the designing of the entire lung generation. Our Foxa2-based CBC methodology diverges from our prior Shh-based CBC model (*Mori et al., 2019*), which targeted the earliest lung precursor lineage, mesendoderm, allowing us to label entire lung epithelium and about 20% of mesenchyme, enough for leading to generating the whole lungs efficiently. Accidentally, we also found that Fgfr2 deficiency in the Foxa2 lineage caused a thymus agenesis phenotype, which was simultaneously rescued by injecting pluripotent stem cells (PSCs) into the blastocysts. Our Foxa2 lineage-based CBC approach results in the highly efficient co-generation of functional lungs and thymus, offering new avenues for exploring the future of autologous lung transplantation.

## Results

### Pdgfrα⁺ lineage during gastrulation gives rise to the entire lung mesenchyme

To determine the origin of LPM and pulmonary endothelium that leads to the whole lung generation, we performed lung mesenchyme precursor lineage-tracing analysis using $Pdgfra^{CreERT2/+}$; $Rosa^{tdTomato/+}$ mice. At E5.5 tamoxifen (Tm) administration, E14.5 lung immunostaining analysis showed that the entire lung mesenchyme was labeled with tdTomato but not in the epithelium (*Figure 1A*). E12.5 morphometric analyses showed an extremely high proportion of lung mesenchyme labeling with tdTomato (*Figure 1B, C*). Among three embryos at E12.5, two showed 100% tdTomato labeling for lung mesenchyme markers, while one showed about 60% labeling in the mesenchyme (*Figure 1C*). Sma⁺ airway smooth muscle cells (86.9% ± 22.7), Pdgfrα⁺ lung mesenchyme (78.7% ± 36.9), and VE-cadherin⁺ endothelium (82.5% ± 30.4) were labeled with tdTomato (*Figure 1C*). These results indicate that the whole lung mesenchyme originates from the Pdgfrα lineage around early-to-mid-streak-stage embryos before LPM formation. In addition, the Pdgfrα lineage labels 20.5% ± 18.3 the lung epithelium (*Figure 1B*, arrows), suggesting that the contribution of the Pdgfrα lineage to the lung endoderm is limited. It indicates that the Pdgfrα lineage faces challenges in rendering the entire lung niches effectively defective across both the endoderm and mesoderm for CBC-mediated lung generation.

### Foxa2 lineage labels Pdgfrα⁺ mesendoderm niche during mouse development

Single-cell RNA-seq (scRNA-seq) analysis using Foxa2-Venus fusion protein reporter mice indicated that the Foxa2 lineage might give rise to LPM and DE (*Scheibner et al., 2021*).

Given that Foxa2 and Pdgfrα are expressed during the conversion from mesendoderm to mesenchyme (*Artus et al., 2010*; *Kopper and Benvenisty, 2012*; *Scheibner et al., 2021*; *Tada et al., 2005*), we used Foxa2-lineage-tracing mice ($Foxa2^{Cre/+}$; $Rosa^{tdTomato/+}$) (*Horn et al., 2012*) to determine whether the Foxa2 lineage would label Mixl1⁺ or Pdgfrα⁺ mesendoderm during gastrulation. We found that Mixl1⁺ within the PS region (*Figure 1D*, yellow dotted line), 87.3% ± 5.26 expressed Pdgfrα protein (*Figure 1E, F*). While the Foxa2 lineage appeared in the distal compartment of Mixl1 weakly positive mesendoderm (*Figure 1E*, arrows) at the anterior primitive streak besides the Foxa2 protein-expressing DE (*Figure 1E*, arrowheads), those Foxa2 lineage-driven tdTomato labeled 27.5% ± 1.15 of mid-streak PS and 18.6% ± 8.93 in the Pdgfrα⁺ mesendoderm of PS around mid-streak-stage embryos (*Figure 1F, G*, arrows). These results suggest that the Foxa2 lineage-labeled Pdgfrα⁺ cells are the distal component of mesendoderm, most likely the derivative of posterior epiblasts since Foxa2 is expressed only in the posterior epiblasts before gastrulation (*Scheibner et al., 2021*). Based on

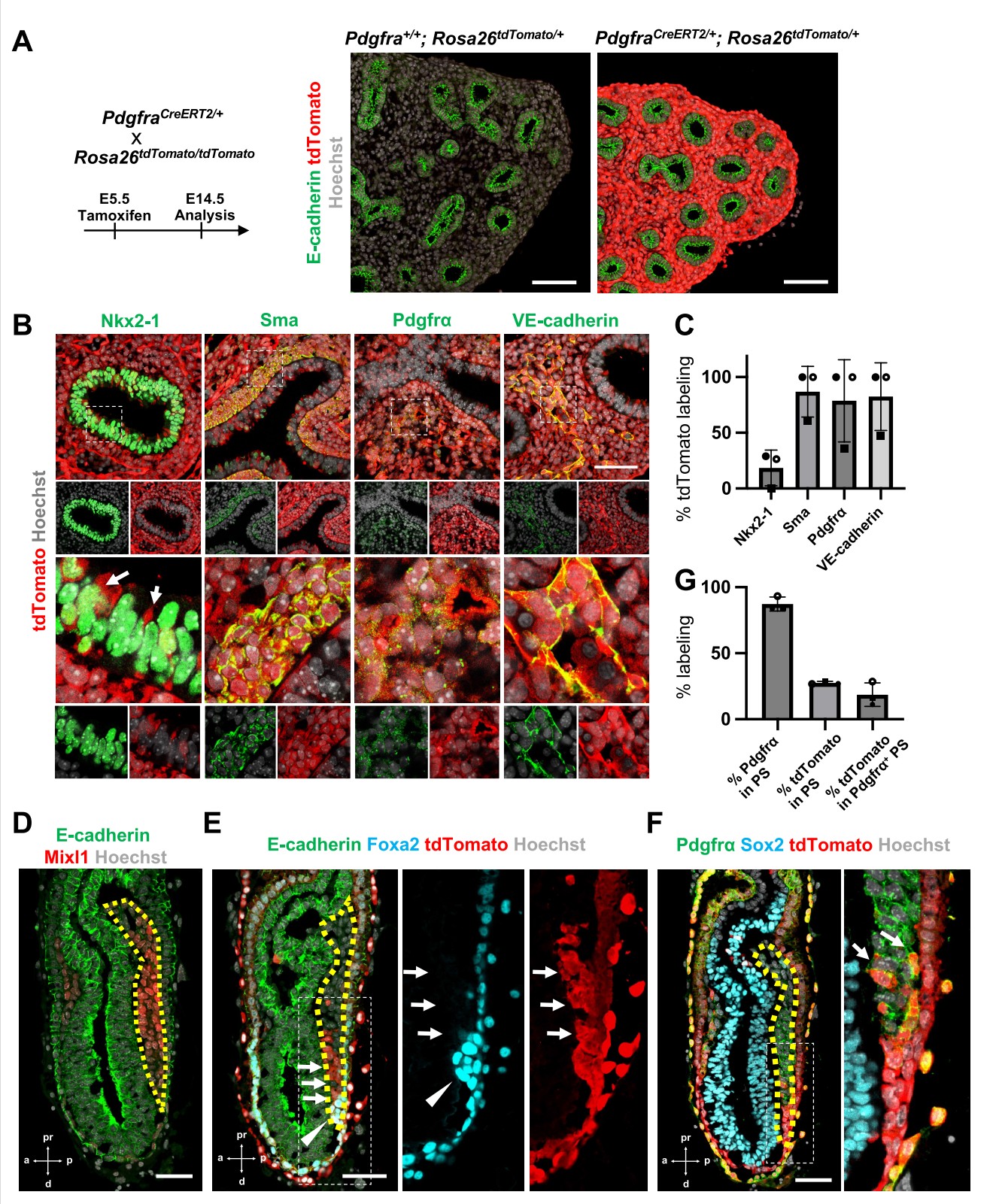

**Figure 1.** Pdgfrα lineage during gastrulation is the origin of the entire pulmonary mesenchyme. (**A**) Left: Schematic of tamoxifen administration. Right: Representative immunofluorescence (IF)-confocal imaging of E14.5 *Pdgfra^CreERT2/+^; Rosa26^tdTomato/+^* lineage tracing mouse lungs. Scale bars = 100 μm. (**B**) Tamoxifen administration at E5.5 and analyzed at E12.5. Pdgfrα-lineage-driven tdTomato (red) labeled the part of lung epithelium (arrows) and entire lung mesenchyme, including Sma⁺ airway smooth muscle cells, Pdgfrα⁺ mesenchyme, and VE-cadherin⁺ capillaries. Enlarged box: dotted box.

*Figure 1 continued on next page*

Figure 1 continued

Scale bar = 50 µm. (**C**) Quantification of *Pdgfra*^CreERT2/+^; *Rosa26*^tdTomato/+^ lineage labeling (*n* = 3 per group, each plot showed different embryos). Error bars represent mean ± standard deviation (SD). (**D–F**) Representative IF-confocal imaging of E6.5 *Foxa2*^Cre/+^; *Rosa26*^tdTomato/+^ embryo. E-cadherin or Sox2 indicates epiblast. (**D**) Mixl1 (red) expression represents primitive streak (PS) (yellow dotted area). (**E**) Foxa2-lineage (red) marked a broader region of the distal component of the anteriolarizing PS (arrows), expressing Foxa2 protein (arrowhead). (**F**) Foxa2-lineage (red)-labeled Pdgfrα (green) expressing mesendoderm (arrows). Scale bars = 50 µm. (**G**) Quantification of Foxa2-lineage labeling in PS (*n* = 3 per group, each plot showed different embryos). Error bars represent mean ± SD.

The online version of this article includes the following source data for figure 1:

**Source data 1.** Quantification of *Pdgfra*^CreERT2/+^; *Rosa26*^tdTomato/+^ lineage labeling by immunofluorescence (IF) morphometric analysis.

**Source data 2.** Quantification of Foxa2-lineage labeling in PS by immunofluorescence (IF) morphometric analysis.

our Foxa2-lineage-tracing data on early embryogenesis, we further examined whether Foxa2 lineage would label lung mesenchyme.

## Foxa2-lineage labeling increased during lung development, leading to occupy the entire lung epithelium and half of the lung mesenchyme, including lung endothelium

Foxa2-lineage-tracing mice (*Foxa2*^Cre/+^; *Rosa*^tdTomato/+^) faithfully target Nkx2-5$^+$ cardiac progenitors associated with the origin of Wnt2$^+$ Isl1$^+$ CPP (*Bardot et al., 2017*; *Peng et al., 2013*) lung mesenchyme. However, there was no conclusive evidence of whether Foxa2 lineage can give rise to Wnt2$^+$ Isl1$^+$ CPP (*Bardot et al., 2017*; *Peng et al., 2013*). Using the Foxa2-lineage-tracing mice, we found from immunostaining that the Foxa2-lineage labeled the entire lung epithelium and most of the lung mesenchyme at E16.5 (*Figure 2A–C*). Quantitative analyses by flow cytometry (FCM) in the E14.5 developing lungs of Foxa2-lineage-tracing mice showed that Foxa2-lineage labeled almost the entire lung epithelium (93.3% ± 2.00) and about 20% of lung mesenchyme (24.2% ± 5.51), including endothelial cells (16.8% ± 5.12) (*Figure 2D, E*), which was a similar proportion of the Foxa2-lineage-labeled Pdgfrα$^+$ mesendoderm (*Figure 1E–G*). Contrary to expectations, Foxa2-lineage-labeled cells increased dramatically throughout lung development (*Figure 2E*). In adulthood, the Foxa2-lineage labeling reached about 99.97% ± 0.06 in lung epithelium, 51.3% ± 1.72 in lung mesenchyme, and 60.0% ± 8.44 in lung endothelial cells, with more than twofold change in the lung endothelium, compared with E14.5 (*Figure 2E*). Morphometric analysis of immunostaining further confirmed the Foxa2-lineage labeling of each mesenchyme marker. Foxa2 lineage marked about 30–40% of E14.5 multiple cell types of lung mesenchyme such as Sma$^+$ smooth muscle cells (31.9% ± 9.16), VE-cadherin$^+$ (39.3% ± 12.5), or Pecam1$^+$ (36.9% ± 10.4) endothelial cells, Pdgfrβ$^+$ pericytes of the pulmonary arteries and pulmonary veins (41.4% ± 10.5), and WT1$^+$ mesothelial cells (31.9% ± 23.7) (*Figure 2F, G*). Interestingly, no clear Foxa2 protein level expression was observed in the mesenchyme of the embryonic lungs at E14.5 and E18.5 (*Figure 2—figure supplement 1A*), consistent with previously reported Foxa2 protein expression patterns (*Wan et al., 2004*). However, the LungMAP deposited single-cell RNA-seq database analysis showed a sporadic Foxa2 transcriptional expression pattern in developing lung mesenchyme, particularly in proliferating endothelium, on E15.5 and E17.5 (*Figure 2—figure supplement 1B*). We confirmed *Foxa2* transcriptional expression in lung mesenchyme by in situ hybridization analysis on E18.5 (*Figure 2—figure supplement 1C, D*). The expression was detected mostly in tdTomato positive but low frequency in the negative cells, which indicates that lung mesenchyme spontaneously expressed *Foxa2* at a transcriptional level and turned on tdTomato during lung development. We also sorted the cell fraction of CD45$^-$CD31$^-$EPCAM$^-$tdTomato$^+$ and CD45$^-$CD31$^-$EPCAM$^-$tdTomato$^-$ from developing lung mesenchyme at E18.5. We observed a slight increase in the relative expression of *Foxa2* in the tdTomato$^+$ fraction of embryonic lung mesenchyme, most likely contributing to the labeling in the Foxa2-lineage-tracing mice (*Figure 2—figure supplement 1E*). These results suggest pulmonary mesenchymal progenitor cells turned on Foxa2 expression slightly at the mRNA level rather than the protein level, which led to a gradual increase in Foxa2 lineage labeling. Furthermore, tdTomato$^+$ endothelial cells were found to have a slightly higher proliferative capacity than tdTomato$^-$ cells (*Figure 2—figure supplement 1*). These results suggest that Foxa2 lineage$^+$ pulmonary mesenchymal appeared during lung development, and lineage labeling gradually increased compared to Foxa2$^-$ lung mesenchyme progenitors throughout lung development.

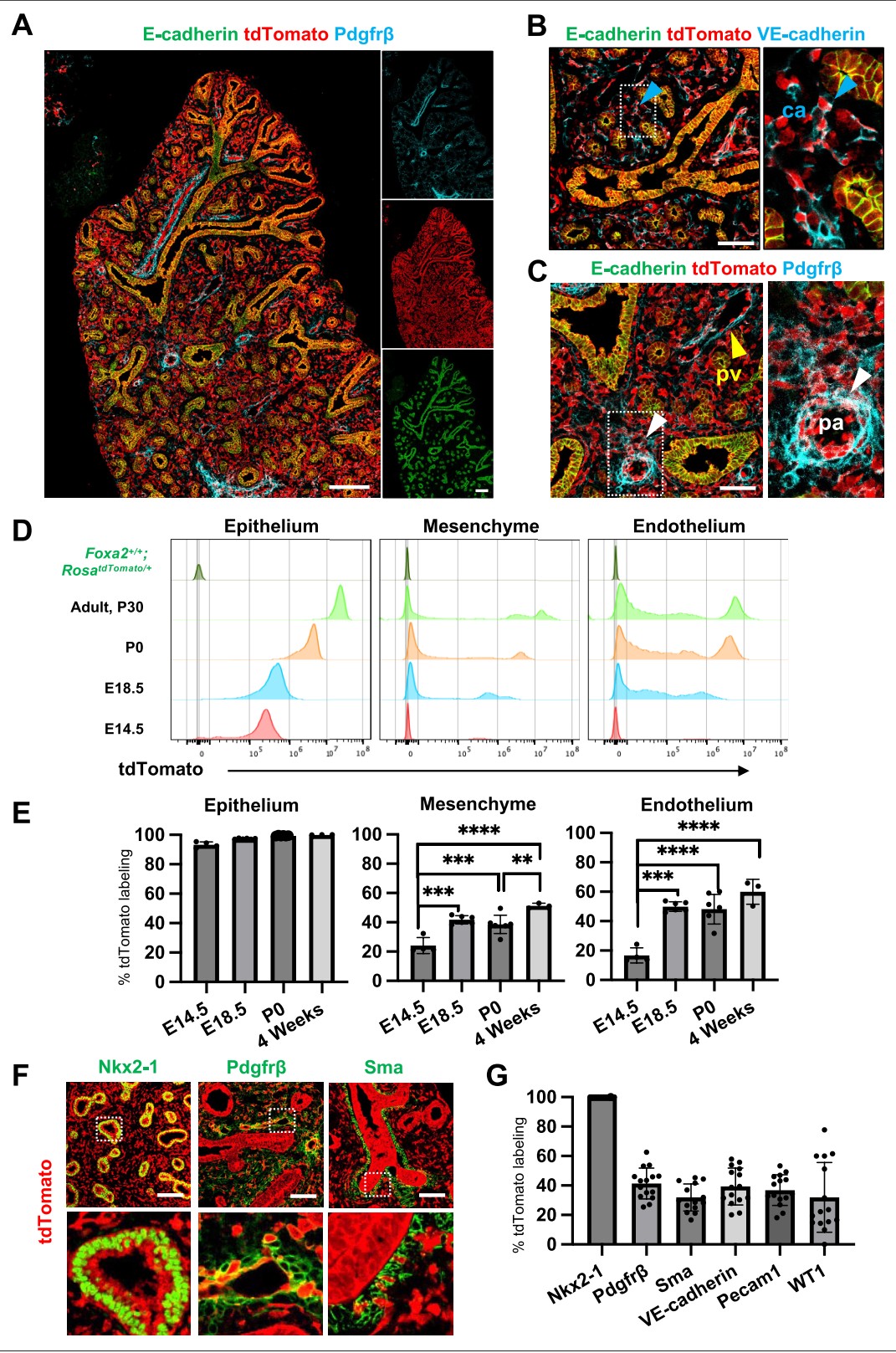

**Figure 2.** Foxa2-lineage gradually increased during lung development and labeled the entire lung epithelium and half of the mesenchyme. (**A–C**) Immunofluorescence (IF)-confocal imaging of E16.5 *Foxa2$^{Cre/+}$; Rosa$^{tdTomato/+}$* embryonic lung: (**A**) Foxa2-lineage (red)-labeled E-cadherin$^{+}$ lung epithelium (green) entirely and Pfgfrβ$^{+}$ mesenchyme (cyan) partially. (**B**) Foxa2-lineage partially labeled VE-cadherin$^{+}$ capillary (ca) (enlarged box, blue

*Figure 2 continued on next page*

*Figure 2 continued*

arrowhead). (**C**) Foxa2-lineage-labeled Pdgfrβ⁺ smooth muscle cells of the pulmonary artery (pa) (enlarged box, white arrowhead) and pulmonary vein (pv, yellow arrowhead). Scale bars (**A**), (**B**), and (**C**) = 200, 100, and 100 µm, respectively. (**D, E**) Representative histograms and the graphs of flow cytometry (FCM) quantitative analyses for CD31⁻Epcam⁺ lung epithelium, CD31⁻Epcam⁻ mesenchyme, and CD31⁺Epcam⁻ endothelium at E14.5, E18.5, P0, and 4 weeks adult ($n$ = 4, 6, 7, and 3, independent biological replicates, respectively) of *Foxa2^{Cre/+}; Rosa^{tdTomato/+}* mouse lungs. The gradual increase of % tdTomato⁺ lineage labeling in both lung mesenchyme and endothelium. Statistical analysis: one-way analysis of variance (ANOVA) with the Tukey post hoc test; statistically significant if **$p < 0.01$, ***$p < 0.001$, ****$p < 0.0001$. Error bars represent mean ± standard deviation (SD). (**F**) Representative IF-confocal imaging of E14.5 *Foxa2^{Cre/+}; Rosa^{tdTomato/+}* embryonic lungs. tdTomato labeled entirely with lung epithelial markers Nkx2-1 (left) but a relatively low proportion of mesenchyme (Pdgfrβ: middle and Sma: right). Scale bar = 50 µm. (**G**) Graphs: The morphometric analysis: % of E14.5 Foxa2-lineage labeling in Nkx2-1⁺ epithelium, Pdgfrβ⁺ mesenchyme, Sma⁺ airway smooth muscle, VE-Cadherin⁺ capillaries, Pecam1⁺ arteries, or WT1⁺ mesothelium ($n$ = 3 per biological replicates, 5 fields per group). Error bars represent mean ± SD.

The online version of this article includes the following source data and figure supplement(s) for figure 2:

**Source data 1.** Flow cytometry (FCM) quantitative analyses for CD31-Epcam⁺ lung epithelium, CD31⁻Epcam⁻ mesenchyme, and CD31⁺Epcam⁻ endothelium at E14.5, E18.5, P0, and 4 weeks adult.

**Source data 2.** The morphometric analysis: % of E14.5 Foxa2-lineage labeling with each lung marker.

**Figure supplement 1.** Foxa2-lineage gradually increased in the mesenchyme and endothelial cells during mouse lung development.

**Figure supplement 1—source data 1.** Foxa2 qPCR for tdTomato⁺ (tdT⁺) or tdTomato⁻ (tdT⁻) CD45⁻CD31⁻EPCAM⁻ mesenchymal cells and CD45⁻CD31⁻EpCAM⁺ epithelial cells.

**Figure supplement 1—source data 2.** Flow cytometry quantitative analyses: EdU labeling cells in CD31⁺Epcam⁻ endothelium at E14.5 and E18.5 of *Foxa2^{Cre/+}; Rosa^{tdTomato/+}* lungs.

## Co-development of endodermal and mesodermal lung progenitors derived from MXIL1⁺ PDGFRα⁺ FOXA2⁺ mesendoderm in the directed differentiation protocol using hiPSC

To determine whether Foxa2 or Pdgfrα mesendoderm is an evolutionarily well-conserved niche that can give rise to both pulmonary endoderm and mesoderm, we modified a previously reported protocol to establish a pulmonary endoderm–mesoderm co-developmentally directed differentiation protocol (*Chen et al., 2017*; *Gotoh et al., 2014*; *Hawkins et al., 2017*; *Huang et al., 2014*; *Konishi et al., 2016*; *Figure 3A*). With this optimized protocol, various hiPSC lines efficiently induced a lung bud-like appearance, indicated by NKX2-1 expression in lung epithelial cells (*Gotoh et al., 2014*; *Konishi et al., 2016*). On day 6, NKX2-1⁺ cells emerged in bud structure (*Figure 3B*, asterisk, and S2A). On day 10, TBX4⁺ lung mesenchyme arose and surrounded the NKX2-1⁺SOX9⁺ lung epithelium (*Figure 3B, C*, *Figure 3—figure supplement 1A, B*). Morphometric analysis of budding structure with lung markers revealed the heterogeneity of bud structures (*Figure 3—figure supplement 1*). The budding morphology appeared 2.8% ±0 .90 per field on day 6 and progressively increased its proportion and reached around 57.4% ± 9.10. Among the developing bud structures, NKX2-1 was initially turned on 44.5% ± 6.41 at day 6, but the proportion progressively increased, and NKX2-1 was positive on the 95.9% ± 3.98 of day 15 budding structures. It indicates that the bud-like structure preceded the lung cell fate specification. Interestingly, we found SOX9 expression in the 27.9% ± 13.1 of NKX2-1+ lung bud cells on day 8, but it dramatically increased and reached nearly 100% positive on day 15 (98.4% ± 1.98). This result indicated that NKX2-1 precedes the expression of SOX9. The proportion of TBX4⁺ cells progressively increased in non-budding areas during the differentiation (48.2% ± 3.17). The qPCR kinetics analyses across the time point further supported the appearance of lung mesenchyme, represented by LPM marker expression peaked on days 6–8: *OSR1*, *FGF10*, *BMP4*, and *PDGFRA* (*Loh et al., 2016*), and smooth muscle cell markers peaked on days 8–10: *ACTA2* and *PDGFRB* (*Han et al., 2020*), and CPP markers peaked on days 8–12; *ISL1*, *WNT2*, *FOXF1*, and *TBX4* (*Kishimoto et al., 2020*; *Peng et al., 2013*), peaked on days 10–14 simultaneously with pulmonary epithelial markers, *NKX2-1* and *CPM* (*Figure 3D*; *Gotoh et al., 2014*).

In this differentiation protocol, NKX2-1⁺ lung endoderm and WNT2⁺TBX4⁺ lung mesoderm were derived from the anteroventral endoderm and mesoderm at day 15 after Activin-mediated DE and LPM induction, respectively (*Chen et al., 2017*; *Huang et al., 2014*). During PS induction from days

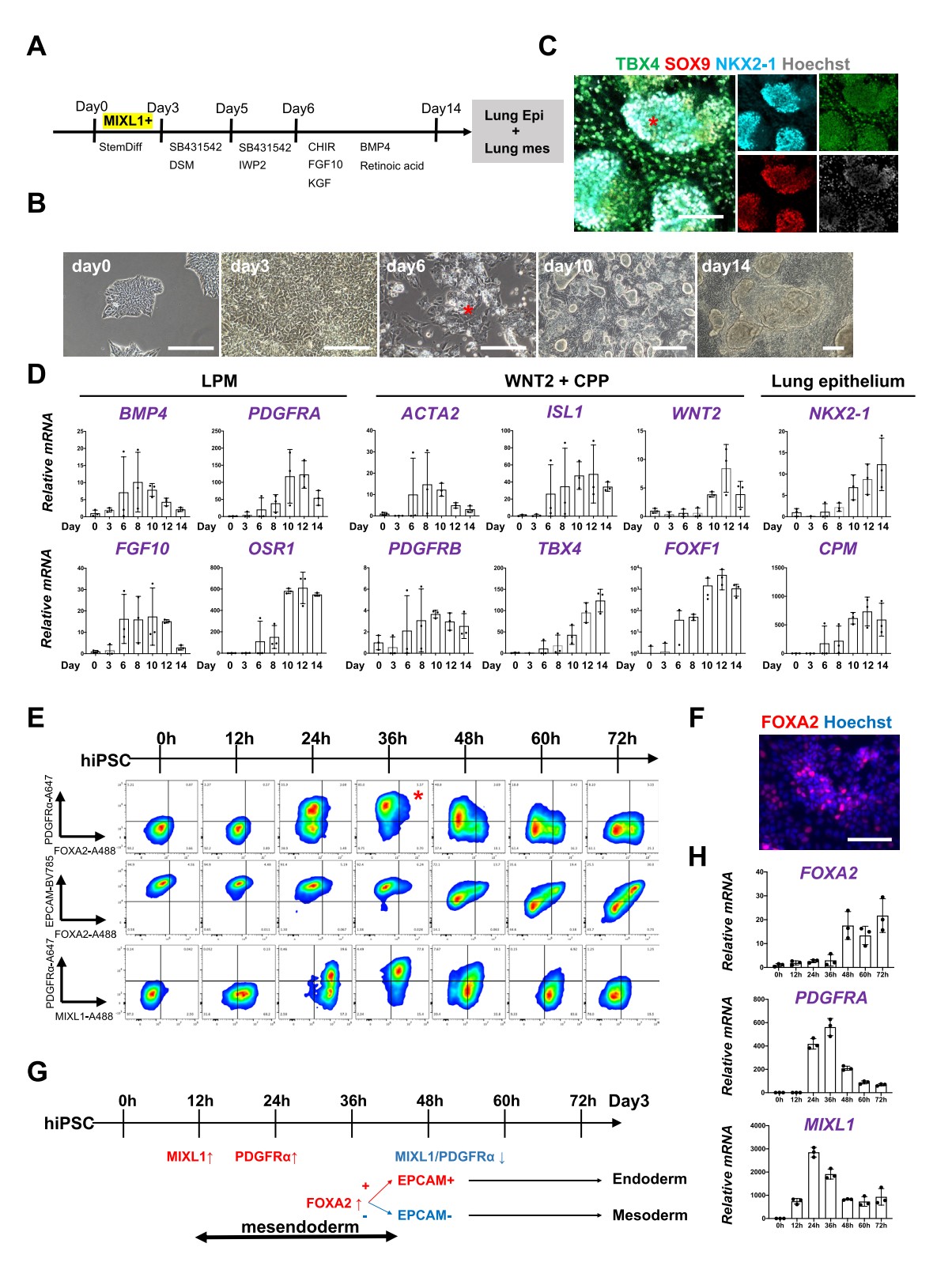

**Figure 3.** Co-development of endodermal and mesodermal lung progenitors derived from MXIL1⁺ PDGFRα⁺ FOXA2⁺ mesendoderm in the directed differentiation protocol using hiPSC. (**A**) Schematic culture protocol of hiPSC-derived endodermal and mesodermal lung progenitor cell co-differentiation. (**B**) Representative phase-contrast images of the directed differentiation time course. Bud structure appeared on day 6 (asterisk). Scale bars = 100 µm. (**C**) Representative immunofluorescence (IF)-confocal imaging of differentiating hiPSCs at day 14 culture. Lung epithelium (NKX2-1),

*Figure 3 continued on next page*

*Figure 3 continued*

distal lung bud epithelium (SOX9), mesenchyme (TBX4), and nucleus (Hoechst) markers. The budding structures expressed SOX9 and NKX2-1 (asterisk), and monolayer cells expressed TBX4. Scale bar = 100 µm. (**D**) RT-PCR analyses of lung mesenchyme and epithelium markers in time course according to the protocol shown in (**A**). Each plot showed a different biological experiment (*n* = 3 independent experiments). Error bars represent mean ± standard deviation (SD). (**E**) Flow cytometry (FCM)-based protein kinetic analyses during definitive endoderm (DE) and lateral plate mesoderm (LPM) induction; MIXL1 expression preceded compared to PDGFRα or FOXA2. FOXA2 appearance in the subset of the PDGFRα⁺ population (red asterisk) (*n* = 3 independent experiments). (**F**) Representative IF imaging of 36-hr-cultured hiPSCs. Scale bar = 100 µm. (**G**) Schematic summary of FCM analysis. (**H**) Quantitative Reverse Transcription Polymerase Chain Reaction (qRT-PCR) analyses further confirmed the preceded *MIXL1* induction and subsequent expression of *PDGFRα* and *FOXA2*. All graphs: Data normalized by undifferentiated induced pluripotent stem cells (iPSCs). Each plot showed a different biological experiment (*n* = 3 independent experiments). Error bars represent mean ± SD.

The online version of this article includes the following source data and figure supplement(s) for figure 3:

**Source data 1.** qRT-PCR analyses of lung mesenchyme and epithelium markers in time course of directed differentiation.

**Source data 2.** qRT-PCR analyses of mesendoderm markers in time course of directed differentiation.

**Figure supplement 1.** Endodermal and mesodermal lung progenitors develop together in the directed differentiation using hiPSC.

**Figure supplement 1—source data 1.** % GFP in CD31⁺Epcam⁻ lung endothelium analyzed by flow cytometry.

**Figure supplement 1—source data 2.** % GFP in CD45⁺ hematopoietic cells analyzed by flow cytometry.

0 to 3, cell surface markers of PDGFRα and EPCAM and intracellular FOXA2 and MIXL1 kinetics were analyzed by FCM every 12 hr (*Figure 3E*). Briefly, 12 hr after the Activin induction, more than 60% of the EPCAM⁺PDGFRα⁻ PS first turned on MIXL1, the mesendoderm marker (*Hart et al., 2002*; *Tada et al., 2005*). Subsequently, the epithelial–mesenchymal transition occurred 24 hr later, as represented by the PDGFRα induction in EPCAM⁺MIXL1⁺ mesendoderm. After 36 hr, more than 90% of MIXL1⁺EPCAM⁺ mesoderm cells expressed PDGFRα. At the same time, expression of FOXA2 appeared in some of those mesoderm cells (*Figure 3E, F*, asterisk). Thereafter, PDGFRα expression decreased, and 72 hr later, mutual FOXA2 induction appeared when EPCAM⁺FOXA2⁺ DE and EPCAM⁻FOXA2⁻ LPM were presented (*Figure 3G*). The dynamics of MIXL1, PDGFRα, and FOXA2 were further revealed by qPCR analysis (*Figure 3H*). These results suggest that PDGFRα⁺ and FOXA2⁺ lineage are redundant but distinct stages of mesendoderm development. It indicates that mesendoderm is the lung precursor niche, with short-time window, contributing to both endodermal and mesodermal-derived lungs, evolutionarily conserved across mouse development and human-directed differentiation.

## Foxa2-driven Fgfr2 conditional knockout showed a lung agenesis phenotype

Since the Foxa2-lineage⁺ mesendoderm forms endodermal and mesodermal lung niches (*Figure 3E*), and the lung mesenchyme lineage was progressively labeled and reached more than 50% by Foxa2 lineage (*Figure 2E*), we investigated the Foxa2-lineage-based CBC strategy. To explore this possibility, we analyzed Foxa2-driven Fgfr2 conditional knockout mouse phenotype (*Foxa2^Cre/+*; *Fgfr2^flox/flox*, hereafter, *Foxa2^Cre/+*; *Fgfr2* cKO). Mitotic signaling via Fgfr2 is required for both lung epithelium and mesenchyme, and systemic knockout mice of Fgf10 or Fgfr2 exhibit a phenotype of lung agenesis (*Arman et al., 1999*; *De Langhe et al., 2006*; *De Moerlooze et al., 2000*; *Sekine et al., 1999*). Based on the results of Foxa2-lineage tracing and the need for Fgfr2 signaling, it was predicted that *Foxa2^Cre/+*; *Fgfr2* cKO mice would be used to generate vacant niches in both lung epithelium and mesenchyme. We did not observe differences in the apparent gross morphology of embryos in *Foxa2^Cre/+*; *Fgfr2* cKO mice however, they exhibited a lung agenesis phenotype and died at birth by respiratory distress (*Figure 4—figure supplement 1A, B*). We did not observe agenesis phenotype in other major endodermal organs, such as the liver, esophagus, pancreas, and intestine (*Figure 4—figure supplement 1*). This is because those organs were Foxa2-lineage labeled (*Figure 4—figure supplement 1*), but Fgfr2 is not critical for their organ formation. We also confirmed that kidney, glandular stomach, hair follicles, tooth bud, and limbs were preserved, which is known to form organ defective phenotype in the Fgfr2 systemic knockout (*Figure 4—figure supplement 1*; *Arman et al., 1999*; *De Langhe et al., 2006*; *De Moerlooze et al., 2000*; *Sekine et al., 1999*), while thymus was found to be the agenesis phenotype in the Foxa2-driven Fgfr2 knockout mice (*Figure 4—figure supplement 2A*).

## Generation of the entire lungs in *Foxa2*-driven *Fgfr2*-deficient mice via CBC

To examine whether donor cells complement the lung agenesis phenotype, we generated nGFP⁺iPSCs from Rosa^{nT-nG} mice (hereafter, nGFP⁺iPSCs) via Sendai virus-mediated reprogramming (*Huang et al., 2014*). nGFP⁺iPSCs were injected into mouse blastocysts (*Figure 4A*), and chimerism was analyzed at E17.5. Strikingly, donor nGFP⁺iPSCs generated whole lungs in *Foxa2^{Cre/+}; Fgfr2* cKO mice, but general chimerism in other organs were diverse (*Figure 4B*, *Figure 4—figure supplement 3A*, and *Table 1*). Importantly, almost the entire lung epithelial, mesenchymal, and endothelial cell population at E17.5 was composed exclusively of nGFP⁺iPSCs (*Figure 4C–E* and *Figure 4—figure supplement 3B–D*). In contrast, wild-type, Shh-driven heterozygous, or knockout mice showed about 6.48–92.3% chimerism in the mesenchymal and endothelial lineages of the lung (*Figure 4E*, *Figure 4—figure supplement 3C*, and *Table 1*). We further compared the chimerism of each cell type in *Foxa2^{Cre/+}; Fgfr2* cKO mice and *Shh^{Cre/+}; Fgfr2* cKO mice. Interestingly, Sma⁺ lung airway mesenchyme (80.5 ± 12.0% vs 46.3 ± 16.6%, p = 0.0057), Pecam⁺ capillary (87.5 ± 7.5% vs 62.7 ± 15.9%, p = 0.0134), and NG2⁺ pericyte (80.6 ± 13.6% vs 57.5 ± 10.6%, p = 0.0171) showed higher chimerism in the complemented lungs of *Foxa2^{Cre/+}; Fgfr2* cKO mice (*Figure 4F–H*, *Figure 4—figure supplement 4*) than that of *Shh^{Cre/+}; Fgfr2* cKO mice. Given that the *Foxa2^{Cre/+}; Fgfr2* cKO mice showed shorter truncated trachea than *Shh^{Cre/+}; Fgfr2* cKO mice (*Figure 4—figure supplement 5A, B*), the rescued trachea of *Foxa2^{Cre/+}; Fgfr2* cKO mice exhibited higher chimerism in epithelium than that of *Shh^{Cre/+}; Fgfr2* cKO mice (95.4 ± 5.72% vs 70.82 ± 18.6%, p = 0 .0223) (*Figure 4—figure supplement 5*). *Foxa2^{Cre/+}; Fgfr2* cKO mice did not show the difference in CD45⁺ cell chimerism (*Figure 4—figure supplement 3*) compared with other genotyping because those cells originate from the yolk sac, liver, or spleen (*Yokomizo et al., 2022*), the off-targeted organs. Interestingly, *Foxa2^{Cre/+}; Fgfr2* cKO mouse showed thymus agenesis phenotype (*Figure 4—figure supplement 2*), but it was rescued by donor PSC injection into the blastocysts of *Foxa2^{Cre/+}; Fgfr2* cKO mice (*Figure 4B*, arrows). The chimerism in the Epcam⁺ thymic epithelium showed higher in *Foxa2^{Cre/+}; Fgfr2* cKO mice than *Shh^{Cre/+}; Fgfr2* cKO mice (92.4 ± 5.10% vs 28.7 ± 25.4%, p = 0 .0006) (*Figure 4—figure supplement 2*). This result suggests that the PSC injection completely rescued both the thymus and lung aplasia phenotype, and lungs and thymus were co-generated in the *Foxa2^{Cre/+}; Fgfr2* cKO mice.

## Fgfr2 knockout in lung mesenchyme suppressed their proliferation and showed higher chimerism of the entire lung by CBC

Previous reports described that Fgfr2 is critical for lung epithelial cell development but also important for lung mesenchyme proliferation (*Yin et al., 2008*). Although tdTomato⁺ Fgfr2 knockout mesenchymal cells remained in early lung development at E14.5 (*Figure 4I*), the percentage of Ki67⁺ proliferating cells was significantly higher in GFP⁺ donor cells compared to tdTomato⁺ host cells (*Figure 4J, K*). These results suggest that generating Fgfr2-deficient niche in the host Foxa2-lineage-derived lung mesenchyme is efficient for donor iPSCs recruitment in the defective Foxa2 lineage mesenchyme niches during development.

## Generation of functional lungs in *Foxa2*-driven *Fgfr2*-deficient mice via CBC

Although we observed nearly the whole lung generation in the *Foxa2^{Cre/+}; Fgfr2* cKO during lung development, it was unclear whether the Foxa2-lineage-based CBC approach was efficient enough for obtaining fully functional lungs and the rescued mice would survive until adulthood. We injected GFP-expressing mouse embryonic stem cells (PSC^{CAG-GFP}) into mouse blastocysts and analyzed the chimeric mice postnatal days to address this issue. We found that *Foxa2^{Cre/+}; Fgfr2* cKO, PSC^{CAG-GFP} rescued mice survived at 4 weeks postnatal days without any difference in their gross appearance and behavior from the littermate control; *Foxa2^{Cre/+}; Fgfr2* hetero, PSC^{CAG-GFP} mice (*Figure 5A*, asterisks). They showed no significant difference in their respiratory functions of *f* (frequency) (232.9 ± 18.5 vs 236.2 ± 25.7%, p = 0.790), TV (tidal volume) (0.253 ± 0.06 vs 0.233 ± 0.06, p = 0.514), Raw (airway resistance) (4.340 ± 1.65 vs 5.393 ± 3.80, p = 0.556), and EF50 (expiratory flow at the point 50% of TV is expired) (2.484 ±0 .513 vs 2.376 ±0 .860, p = 0.793) (*Figure 5B*). Macroscopic analysis showed strong GFP signals but little host-derived tdTomato signals in the lungs of *Foxa2^{Cre/+}; Fgfr2* cKO, PSC^{CAG-GFP} rescued mice compared to littermate controls (*Figure 5C, D*), suggesting efficient donor

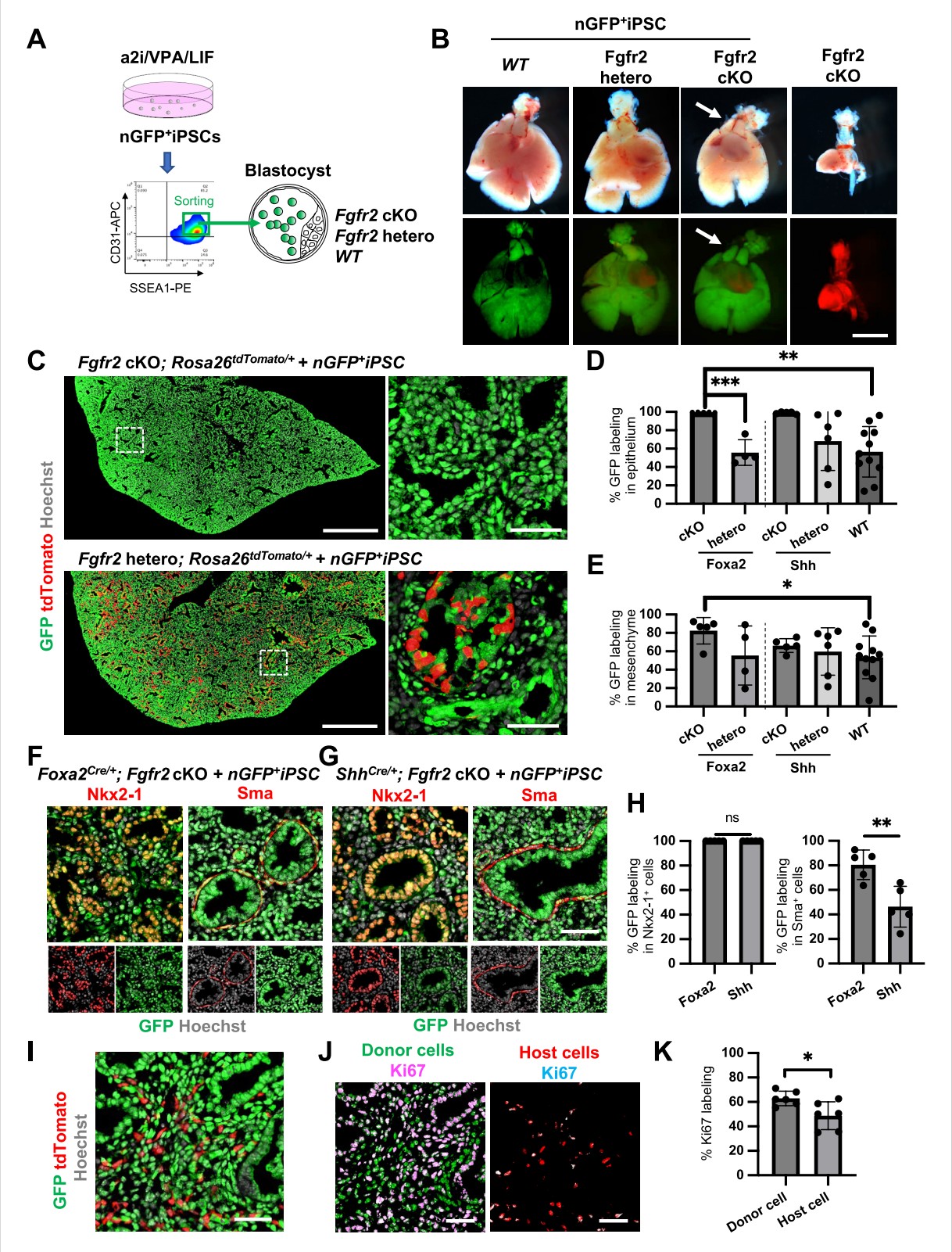

**Figure 4.** Generation of the entire lungs in *Foxa2*-driven *Fgfr2*-deficient mice via conditional blastocyst complementation (CBC) at E17.5. (**A**) Schema of CBC experiment: a2i/VPA/LIF-treated SSEA1[high] CD31[high] nGFP+iPSCs were sorted and injected into WT, *Fgfr2* hetero (heterozygous: *Foxa2*[cre/+]; *Fgfr2*[flox/+]; *Rosa26*[tdTomato/+]), and *Fgfr2* cKO (homozygous: *Foxa2*[cre/+]; *Fgfr*[flox/flox]; *Rosa26*[tdTomato/+]) blastocysts. (**B**) Gross morphology, GFP (green: donor nGFP+iPSCs-derived signals) and tdTomato (host Foxa2-lineage-derived signals) fluorescence of freshly isolated lungs from E17.5 chimeric *WT* (left),

*Figure 4 continued on next page*

*Figure 4 continued*

*Fgfr2* hetero (left middle) and *Fgfr2* cKO (right middle) that were injected with nGFP⁺iPSCs. Control: littermate *Fgfr2* cKO mouse without nGFP⁺iPSCs injection (right). Arrows: rescued thymus formation. (**C**) Representative immunofluorescence (IF)-confocal imaging of E17.5 *Fgfr2* cKO or *Fgfr2* hetero lungs injected with nGFP⁺iPSCs. Dotted lines: enlarged images: *Fgfr2* hetero lungs holding host-derived cells. On the other hand, *Fgfr2* cKO lungs were entirely composed of donor-derived nGFP⁺ cells. Scale bars left and right = 0.5 mm and 50 μm. Graphs: % GFP in (**D**) CD31⁻Epcam⁺ lung epithelium and (**E**) CD31⁻Epcam⁻ mesenchyme analyzed by flow cytometry. Each plot showed different biological animals. *Foxa2^{Cre}; Fgfr2* cKO; *Rosa^{tdTomato}* (n = 5, independent biological replicates), *Foxa2^{Cre}; Fgfr2* hetero; *Rosa^{tdTomato}* (n = 4), *Shh^{Cre}; Fgfr2* cKO; *Rosa^{tdTomato}* (n = 5), *Shh^{Cre/+}; Fgfr2* hetero; *Rosa^{tdTomato}* (n = 6), and *WT* (n = 11). Statistical analyses: unpaired Student's *t*-test, significance at *p < 0.05, **p < 0.01, ***p < 0.001. Error bars represent mean ± standard deviation (SD). Representative IF-confocal imaging of (**F**) *Foxa2^{Cre}; Fgfr2* cKO; *Rosa^{tdTomato}* and (**G**) *Shh^{Cre}; Fgfr2* cKO; *Rosa^{tdTomato}* injected with nGFP⁺iPSCs. (**H**) Morphometric analysis: % GFP in Nkx2-1+ epithelial cells (left) and Sma⁺ airway smooth muscle cells (right) in *Foxa2^{Cre}; Fgfr2* cKO; *Rosa^{tdTomato}* and *Shh^{Cre}; Fgfr2* cKO; *Rosa^{tdTomato}*. Statistical analyses: unpaired Student's *t*-test, significance if *p < 0.05, **p < 0.01, ns: non-significant. Error bars represent mean ± SD. Scale bars = 50 μm. (**I**) Representative IF-confocal imaging of E14.5 lung of *Fgfr2* cKO injected with nGFP⁺iPSCs. GFP and tdTomato indicate donor and host-derived cells, respectively. Scale bar = 20 μm. (**J**) Split images of (I) visualizing GFP⁺ donor cells and tdTomato⁺ host cells, co-stained with Ki67. Scale bar = 20 μm. (**K**) Graphs: % Ki67 labeling in mesenchymal cells of E14.5 *Foxa2^{Cre/+}; Fgfr2* cKO chimeric lungs. Statistical analyses: paired Student's *t*-test, significance if *p < 0.05, ns: non-significant. Error bars represent mean ± SD.

The online version of this article includes the following source data and figure supplement(s) for figure 4:

**Source data 1.** Morphometric analysis: % GFP in Nkx2-1⁺ epithelial cells and Sma⁺ airway smooth muscle cells in *Foxa2^{Cre}; Fgfr2* cKO; *Rosa^{tdTomato}* and *Shh^{Cre}; Fgfr2* cKO; *Rosa^{tdTomato}*.

**Source data 2.** Morphometric analysis: % Ki67 labeling in mesenchymal cells of E14.5 *Foxa2^{Cre/+}; Fgfr2* cKO chimeric lungs.

**Figure supplement 1.** Lung agenesis phenotype in the *Foxa2^{Cre/+}Fgfr2* cKO mice.

**Figure supplement 2.** The complemented embryos of *Foxa2^{Cre/+}; Fgfr2* cKO *and Shh^{Cre/+}; Fgfr2* cKO + nGFP⁺iPSCs showed different chimerism in thymus epithelial cells.

**Figure supplement 2—source data 1.** Morphometric immunofluorescence (IF) analysis of chimeric thymus: % chimerism of thymus epithelium and mesenchyme.

**Figure supplement 3.** The complemented embryos of *Foxa2^{Cre/+}; Fgfr2* cKO + nGFP⁺iPSCs showed normal gross morphology and various chimerism.

**Figure supplement 4.** The complemented embryos of *Foxa2^{Cre/+}; Fgfr2* cKO *and Shh^{Cre/+}; Fgfr2* cKO + nGFP⁺iPSCs showed different chimerism in each lung cell type.

**Figure supplement 4—source data 1.** Morphometric analysis of E17.5 chimeric lungs in each lung mesenchyme marker.

**Figure supplement 5.** The complemented embryos of *Foxa2^{Cre/+}; Fgfr2* cKO *and Shh^{Cre/+}; Fgfr2* cKO + nGFP⁺iPSCs showed different chimerism in tracheal epithelial cells.

**Figure supplement 5—source data 1.** Morphometric immunofluorescence (IF) analysis of the chimeric trachea: % chimerism of tracheal epithelium and mesenchyme.

complementation in both epithelial and mesenchymal lung progenitors. FCM analyses confirmed a significantly lower proportion of the residual tdTomato⁺ host-derived cells in the epithelium (0.29% ± 0.40 vs 45.9% ± 24.1, p = 0.0004), mesenchyme (1.80% ± 1.94 vs 28.1% ± 11.5, p < 0.0001), and endothelium (3.86% ± 4.89 vs 44.6% ± 16.5, p < 0.0001) of the complemented adult lungs compared to the littermate control (**Figure 5E, F**). IF of lungs from *Foxa2^{Cre/+}; Fgfr2* cKO, PSC^{CAG-GFP} rescued animals confirmed extensive GFP overlap with markers of the airway and alveolar epithelial cell types and mesenchyme (**Figure 5G**). We also confirmed that PSC^{CAG-GFP} injection into *Foxa2^{Cre/+}; Fgfr2* cKO blastocysts simultaneously rescued the thymus agenesis phenotype (**Figure 5C, H**, arrows).

These results indicate that *Foxa2^{Cre/+}; Fgfr2* cKO provides defective niches in both lung epithelium and mesenchyme and thymic epithelium to be complemented by donor PSCs, enabling the entire lungs and thymus co-generation.

## Discussion

The presence of committed organ precursors capable of contributing to multi-embryonic layers after pluripotent epiblast formation has been assumed, but the identity and origin of the lung precursors needed to be better defined. Although the Foxa2 lineage could not give rise to the entire lung cells, we have identified a Foxa2 lineage that provides a significant steppingstone for facilitating functional whole lung generation. Since donor cells formed about 50–80% chimerism in the complemented lung mesenchyme niches at E14.5 (**Table 2**) and endogenous Foxa2 lineage forms only 20–30% in the lung mesenchyme at E14.5, Fgfr2 defects at the Foxa2 lineage in early lung development are important but not sufficient for the complete lung mesenchyme niche complementation. Losing Fgfr2 expression in

**Table 1.** E17.5 chimerism of Foxa2 or Shh promoter-driven conditional blastocyst complementation.

| | | Liver | Hematopoietic cells | Lung Whole | Epithelium | Mesenchyme | Endothelium |
|---|---|---|---|---|---|---|---|
| Foxa2-Cre | Fgfr2 cKO | 56.2 | 71.3 | 91.8 | 99.8 | 92.3 | 73.1 |
| | | 54.5 | 64.9 | 90.6 | 99.7 | 89.8 | 91.7 |
| | | 48.3 | 62.9 | 90.3 | 99.6 | 86.7 | 76.7 |
| | | 39.4 | 57.8 | 87.7 | 99.8 | 86.2 | 73.9 |
| | | 14.4 | 21.1 | 58.9 | 99.5 | 57 | 38.6 |
| | Fgfr2 hetero | 60.2 | 68.7 | 65.9 | 51.6 | 75 | 68.1 |
| | | 55.8 | 70 | 88.8 | 76.1 | 89 | 88.5 |
| | | 24.8 | 32 | 42.1 | 44.6 | 38.4 | 41.1 |
| | | 9.7 | 12.6 | 21.2 | 50.7 | 19.3 | 13.4 |
| Shh-Cre | Fgfr2 cKO | 35.7 | 38.9 | 73.6 | 99.7 | 70.2 | 69.3 |
| | | 13 | 18.8 | 79.2 | 99.9 | 75.3 | 72.6 |
| | | 28.8 | 26.6 | 68.3 | 97.3 | 66 | 47.3 |
| | | 43.1 | 53.8 | 67.5 | 98 | 64.8 | 54.9 |
| | | 19.2 | 18.9 | 63.7 | 99.5 | 55.1 | 60.1 |
| | Fgfr2 hetero | 57.4 | 12.7 | 33.6 | 38.3 | 33.4 | 22.1 |
| | | 44.4 | 44.8 | 84.3 | 96.8 | 81.8 | 71.2 |
| | | 13.8 | 52.8 | 83.1 | 98.3 | 79.1 | 82.9 |
| | | 16.2 | 6.8 | 26.1 | 20.9 | 21.6 | 9.58 |
| | | 41.2 | 48.6 | 70.3 | 83.6 | 66.3 | 52.6 |
| | Wild type | 40 | 41.5 | 76.8 | 71 | 76.5 | 66.6 |
| | | 79.1 | 100 | 89.5 | 90.4 | 83.2 | 80.7 |
| | | 60.1 | 53.8 | 68.5 | 77.6 | 65.3 | 66.4 |
| | | 22.3 | 17.8 | 56.7 | 17.8 | 60.8 | 65.6 |
| | | 20.1 | 15.4 | 53.1 | 54.5 | 52 | 50.8 |
| | | 2.64 | 4.08 | 8.82 | 13.8 | 6.48 | 8.69 |
| | | 24.3 | 19.1 | 54.9 | 65.1 | 51.1 | 30.4 |
| | | 23.9 | 18.6 | 54.2 | 44 | 55.3 | 30.9 |
| | | 49 | 51 | 59.6 | 76.7 | 56.5 | 31.3 |
| | | 19.8 | 15.6 | 45.7 | 47.4 | 30.4 | 70.4 |
| | | 2.9 | 2.19 | 43.3 | 39.4 | 37 | 32.5 |
| | | 77.1 | 69.7 | 91.3 | 96.2 | 89.7 | 91.3 |

Foxa2 lineage cells resulted in lower proliferative ability than donor cells during chimeric lung mesenchyme development (*Figure 4I–K*), allowing progressive lung mesenchyme complementation in lung development and postnatal days, leading to promoting efficient whole lung generation.

Lung epithelial cell precursors were well known to be Shh$^+$ DE in the lung development field (*Cardoso and Kotton, 2008*; *Christodoulou et al., 2011*; *Harris et al., 2006*; *Kadzik and Morrisey, 2012*; *Tian et al., 2011*; *Weaver et al., 1999*; *Xing et al., 2008*), and indeed, the Shh-lineage traces putative DE-derived epithelial lineage but little lung mesenchyme (*Figure 4—figure supplement 3*). Targeting the endodermal lung lineage driven by Shh was sufficient for lung epithelial complementation but insufficient to generate whole lungs, and host-derived cells remained substantially in the mesodermal lung component (*Mori et al., 2019*).

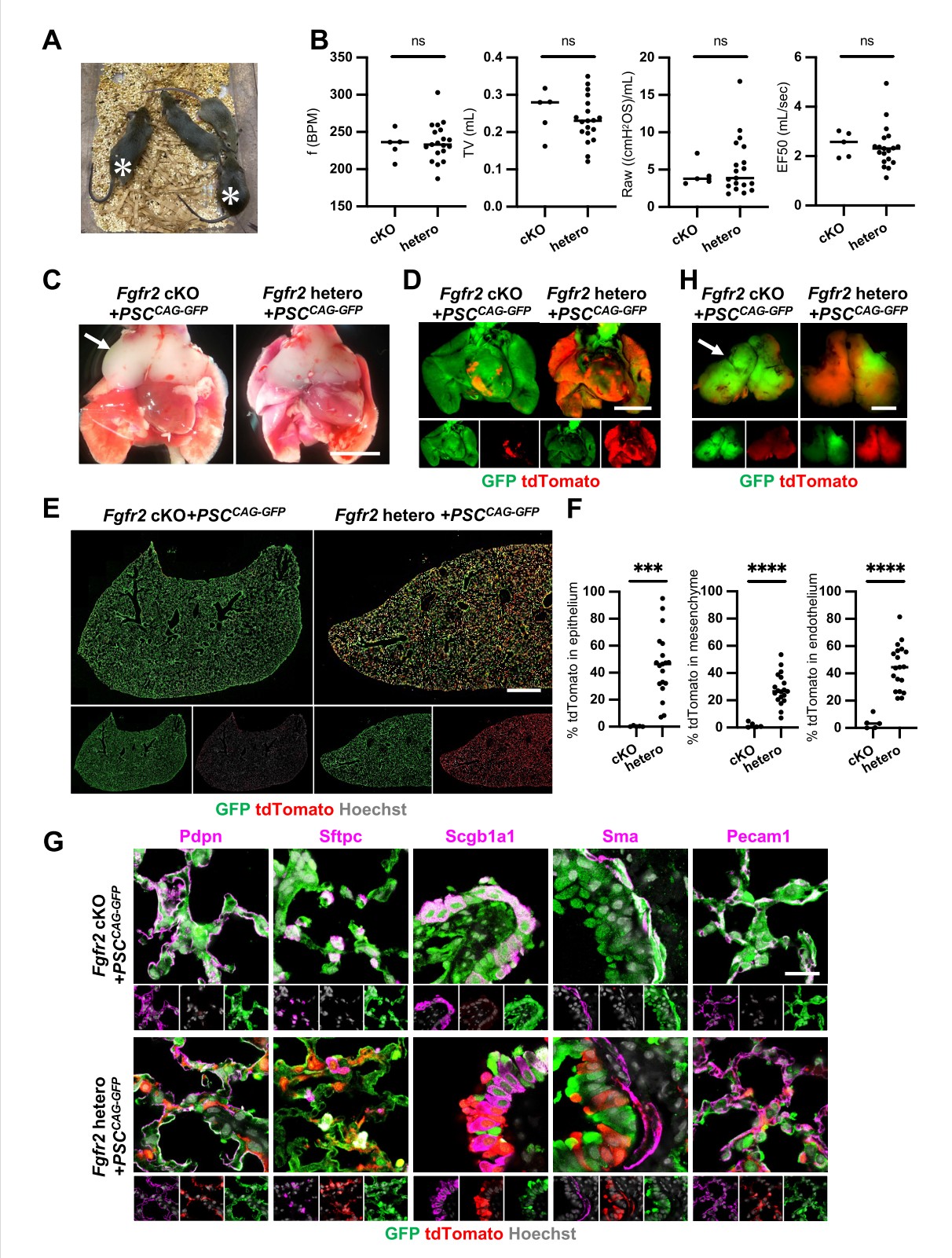

**Figure 5.** Generation of fully functional adult lungs in *Foxa2*-driven *Fgfr2*-deficient mice via conditional blastocyst complementation (CBC). (**A**) Adult mice of *Foxa2*^Cre/+^; *Fgfr2* cKO; *Rosa*^tdTomato^ (asterisks) or *Foxa2*^Cre/+^; *Fgfr2* hetero; *Rosa*^tdTomato^ injected with PSC^CAG-GFP^. (**B**) Graphs: Respiratory function analysis of *f* (frequency), TV (tidal volume), Raw (airway resistance), and EF50 (expiratory flow at the point 50% of TV is expired) of *Foxa2*^Cre/+^; *Fgfr2* cKO; *Rosa*^tdTomato^ (*n* = 5, independent biological replicates), *Foxa2*^Cre/+^; *Fgfr2* hetero; *Rosa*^tdTomato^ (*n* = 19). Statistical analyses: unpaired Student's *t*-test,

*Figure 5 continued on next page*

*Figure 5 continued*

significance at ns: non-significant. (**C**) Gross morphology of 4 weeks rescued lung and thymus from *Foxa2^Cre/+^; Fgfr2* cKO; *Rosa^tdTomato^* (*Fgfr2* cKO) or *Foxa2^Cre/+^; Fgfr2* hetero; *Rosa^tdTomato^* (*Fgfr2* hetero) injected with PSC^CAG-GFP^. Scale bar = 5 mm. (**D**) Fluorescent signals of 4 weeks chimeric lungs. GFP (green: donor PSC^CAG-GFP^-derived signals) and tdTomato (red: host Foxa2-lineage-derived signals) fluorescence of freshly isolated lungs after removing thymus. Scale bar = 5 mm. (**E**) Representative immunofluorescence (IF) tiled-scan confocal-IF imaging of 4 weeks *Fgfr2* cKO or *Fgfr2* hetero lungs injected with PSC^CAG-GFP^. *Fgfr2* cKO lungs showed less host origin tdTomato⁺ cells compared to *Fgfr2* hetero. Scale bar = 3 mm. (**F**) Flow cytometry (FCM) analysis: Graphs: Remaining tdTomato⁺ cell population of each cell type in 4 weeks *Fgfr2* cKO or *Fgfr2* hetero lungs injected with PSC^CAG-GFP^. Statistical analyses: unpaired Student's *t*-test, significance if ***$p < 0.001$ and ****$p < 0.0001$. (**G**) Representative IF-confocal imaging of 4 weeks chimeric lungs from *Fgfr2* cKO or *Fgfr2* hetero injected with PSC^CAG-GFP^ expressed differentiated cell markers such as Pdpn (alveolar type1 cells), Sftpc (type 2 cells), Scgb1a1 (club cells), Sma (smooth muscle cells), and VE-cadherin (endothelial cells). Scale bar = 20 μm. (**H**) Fluorescent signals of 4 weeks chimeric thymus. Arrows in C and H: rescued thymus formation, Scale bar = 1 mm.

The online version of this article includes the following source data and figure supplement(s) for figure 5:

**Source data 1.** Respiratory function analysis of *f* (frequency), TV (tidal volume), Raw (airway resistance), and EF50 (expiratory flow at the point 50% of TV is expired) of *Foxa2^Cre/+^; Fgfr2* cKO; *Rosa^tdTomato^*.

**Source data 2.** Flow cytometry (FCM) analysis: remaining tdTomato⁺ cell population of each cell type in 4 weeks *Fgfr2* cKO or *Fgfr2* hetero lungs injected with PSC^CAG-GFP^.

**Figure supplement 1.** Summary of the results and proposed models.

In contrast to lung epithelial precursors, the orderly commitment and the origin of the entire pulmonary mesenchyme needed to be better defined. We showed that gastrulating Pdgfrα lineage is the origin of the whole lung mesoderm, including endothelium (*Figure 1A, B*). Furthermore, Foxa2 or Pdgfrα lineage labels a population primarily comprised of the earliest specified precursors in the distal compartment of mesendoderm, similar to human iPSC cell-derived directed differentiation protocol. Our findings, summarized in *Figure 5—figure supplement 1A*, pinpoint the lineage hierarchy of specified lung precursors in gastrulating mesendoderm, further supported by the scRNA-seq analysis in early embryonic development (*Figure 5—figure supplement 1*; *Pijuan-Sala et al., 2019*).

As previously indicated that nascent mesoderm differentiation into a CPP fate (*Bardot et al., 2017*; *Devine et al., 2014*; *Ng et al., 2022*; *Peng et al., 2013*), we clarified the orderly mesendoderm progression of gene expression, Mixl1, Pdgfrα, and Foxa2, and lung progenitor-related markers that parallels the commitment of Foxa2⁺ Pdgfrα⁺ mesendoderm to an LPM and DE fate in the lung-directed differentiation protocol using human iPSC (*Figure 3*), not clarified in the previous pioneering works (*Chen et al., 2017*; *Hawkins et al., 2017*; *Huang et al., 2014*). We uncovered that PDGFRα⁺ mesendoderm primes the cell fate of Foxa2⁺ DE after the MIXL1⁺ mesendoderm specification in the directed

**Table 2.** E14.5 chimerism of Foxa2 promoter-driven conditional blastocyst complementation.

| | | Liver | Lung | | | |
| | | | Whole | Epithelium | Mesenchyme | Endothelium |
|---|---|---|---|---|---|---|
| **Foxa2-Cre** | *Fgfr2* cKO | 59.9 | 81.9 | 99.4 | 81.4 | 86.2 |
| | | 51 | 75.4 | 99.4 | 75.4 | 56.8 |
| | | 25.3 | 56.5 | 97.4 | 54.1 | 63.2 |
| | *Fgfr2* hetero | 37.1 | 54.4 | 56.9 | 54 | 58.9 |
| | | 29.8 | 48.6 | 83.3 | 47.7 | 47.3 |
| | | 18.7 | 43.4 | 18 | 45 | 31 |
| | | 0.54 | 0.24 | 0.91 | 0.1 | 2.11 |
| | | 0.4 | 0.17 | 1.02 | 0.058 | 1.23 |
| | Wild type | 39 | 68.7 | 56.2 | 69.2 | 67.5 |
| | | 29.4 | 81.1 | 62.7 | 81.8 | 80.8 |
| | | 28.6 | 73.9 | 39.6 | 75.6 | 76.4 |
| | | 13.8 | 29.5 | 5.57 | 30.4 | 29.6 |
| | | 0 | 2.22 | 5.52 | 1.5 | 16 |

differentiation. How the lung progenitors acquire the lung epithelial cell fate of NKX2-1 with the distal lung tip marker, SOX9, and TBX4$^+$ mesenchyme marker is also fundamental for understanding human lung development.

Our lineage tracing analysis also highlighted the unanticipated Foxa2 lineage program, the progressive increase of lineage labeling by spontaneous expression of *Foxa2 mRNA* that occupies more than half of the lung mesenchyme during lung development. Foxa2 lineage$^+$ lung mesenchyme is a part of Pdgfrα lineage$^+$ cells, potentially providing a unique competitive developmental niche during lung development. Further analyses using Foxa2CreERT2-lineage-tracing mice are required to clarify it. Intriguingly, the Foxa2 lineage$^+$ mesenchyme did not show any discrete fates that would predict anatomical localization—depleting Fgfr2 in the Foxa2 lineage results in the loss of the Fgfr2 mitogen-mediated function of Foxa2 lineage$^+$ lung mesenchyme, which leads to the loss of proliferative ability in most host lung mesenchyme. The Foxa2-lineage labeling in lung mesenchyme is around 20% at E14.5 (*Figure 2D, E*). Moreover, the donor cell complementation in lung mesenchyme at E14.5 still holds host-derived lung mesenchyme but decreases at E17.5 and adulthood (*Figures 4I, C, 5E, F*). These results indicate that the defective Foxa2 lineage is critically essential for efficient lung complementation.

We also found that the combination of Fgfr2 depletion in Foxa2 lineage preserved other major organs, such as kidney, glandular stomach, hair follicles, tooth bud, and limbs, distinct from the agenesis or dysgenesis phenotype caused by the systemic Fgfr2 depletion (*Arman et al., 1999*; *De Langhe et al., 2006*; *De Moerlooze et al., 2000*; *Sekine et al., 1999*). This is because the timing and the requirement of Fgfr2 are different from the lineage in each organ (*Bates, 2011*; *Revest et al., 2001*). We accidentally found a thymus agenesis phenotype in the *Foxa2$^{Cre/+}$; Fgfr2* cKO mice but not in *Shh$^{Cre/+}$; Fgfr2* cKO mice and its complementation. The thymus is raised in the third pharyngeal pouch with the requirement of the Forkhead Box N1 (Foxn1) transcription factor in epithelial cells (*Chhatta et al., 2021*), while the redundant role of Foxa2 and Foxa1 in T cell maturation was reported (*Ramachandran et al., 2022*). The previous report generated thymus using BC by the Foxn1 depletion in host animals, followed by the PSC injections (*Yamazaki et al., 2022*). Our reports showed novel evidence of the requirement of Fgfr2 signaling for thymus formation, specifically in the Foxa2 lineage, associated with the previous statements regarding the critical role of Fgfr2 in thymus formation and thymic epithelial cell differentiation (*Dooley et al., 2007*; *Revest et al., 2001*). This study showed evidence that thymus formation is essentially formed in the Foxa2 lineage but not Shh, supported by the previous report that the dynamic Shh expression regulates larynx-esophageal separation and esophageal constriction on the fourth pharyngeal pouches above the lungs during development (*Ramachandran et al., 2022*). Corresponding to the previous report, the mice with the rescued thymus showed no apparent signs of immune deficiency and survived until adulthood (*Yamazaki et al., 2022*). In these complemented mice, immune cells were most likely educated in self-recognition by donor cells' derived thymus epithelium and induced central tolerance. How thymic epithelium is distinctively lineage labeled by Foxa2 and Shh, and the tolerance difference should be investigated further. It is also important to note that GvHD is a crucial risk factor for long-term survival following human lung transplantation (*Bos et al., 2022*). If we transplant the complemented lungs as a graft containing the conditioned host- and donor-derived immune cells to the recipient that provided donor PSC, it may not cause GvHD. This unique research direction of autologous lung transplantation must be extensively investigated using recipient-derived iPSC with our CBC lung generation models.

An unambiguous proof of the developmental origins of patterned organs is critical for developing regenerative strategies and a better understanding of the genes responsible for congenital malformations. The human-derived developmental program modification to match the host animals is vital for the future development of whole lung generation via CBC using human iPSCs. More broadly, our studies offer a new paradigm that can be applied to modeling various congenital lung diseases of both lung mesenchyme and epithelium and future autologous transplantation therapies using iPSC in the near future.

## Methods

### Mouse

*Shh*$^{Cre/+}$ mice (cat. 05622), *Rosa26*$^{tdTomato/ tdTomato}$ mice (cat. 07914), Rosa26$^{nT-nG/nT-nG}$ mice (cat. 023035), and *Pdgfra*$^{CreERT2/+}$ mice (cat. 032770) were obtained from the Jackson Lab. X. Zhang kindly gifted *Fgfr2*$^{flox/flox}$ mice. We further backcrossed these mice for over three generations with CD-1 mice (cat. 022) from the Charles River. Dr. Nicole C Dubois kindly provided *Foxa2*$^{Cre/Cre}$ mice (129 × B6 mixed background). For conditional deletion of *Fgfr2* (*Fgfr2* cKO), we crossed *Fgfr2*$^{flox/flox}$; *Rosa26*$^{tdTomato/tdTomato}$ females with *Foxa2*$^{Cre/Cre}$; *Fgfr2*$^{flox/+}$, *Foxa2*$^{Cre/+}$; *Fgfr2*$^{flox/+}$, or *Shh*$^{Cre/+}$; *Fgfr2*$^{flox/+}$ males, respectively. PCR performed genotyping of the *Shh-Cre*, *Rosa26-tdTomato*, *Rosa26-nTnG*, and *Pdgfra-CreERT2* alleles according to the protocol provided by the vendor. For detecting the *Fgfr2* floxed allele, we performed PCR using the primer sets: 5'-ATAGGAGCAACAGGCGG-3', and 5'-CAAGAGGCGACC AGTCA-3' (*Mori et al., 2019*). For the CBC, genotyping of chimeric animals was confirmed by 10,000 sorted cells from GFP-negative, tdTomato-positive sorted liver or lung cells. Briefly, liver or lung fibroblasts of adult chimeric mice were primarily cultured in Dulbecco's modified Eagle's medium (DMEM) + 10% fetal bovine serum (FBS) supplemented with penicillin/streptomycin. Then tdTomato$^+$ cells were sorted out, and genotyping was confirmed. In addition to this, IF imaging of Fgfr2 in tdTomato$^+$ tracheal epithelial cells was also conducted to confirm Fgfr2 knockout for adult chimera analysis using tyramid signal amplification (AKOYA Biosciences, SAT705A001EA). For lineage tracing with *Pdgfra-*$^{CreERT2/+}$; *Rosa26*$^{tdTomato/+}$ mice, 1 dose of 200 µg tamoxifen (MedChem Express, HY-13757A) per g of body weight was given via oral gavage injection. All animal experiments were approved by Columbia University Institutional Animal Care and Use Committee in accordance with US National Institutes of Health guidelines.

### Culture of human PSCs

All PSC lines were maintained in feeder-free conditions on laminin iMatrix-511 silk E8 (Amsbio, AMS.892021) in StemFit 04 complete Medium (Amsbio, SFB-504), supplemented with Primocin (Invivogen, ant-pm-1), and passaged with TrypLE Select (Gibco, A1285901). All human iPSC lines used were characterized for pluripotency and were found to be karyotypically normal. The BU3NGST cell line was kindly gifted by Dr. Finn Hawkins and Dr. Darrell Kotton at Boston University, Boston, MA. Dr. Jennifer Davis, the University of Washington School of Medicine, Seattle, WA, kindly gifted the Rainbow cell line. PD2 and TD1 hiPSC were generated from deidentified commercially available human peripheral blood mononuclear cell and tracheal epithelial cell lines via the manufacturing protocol of Sendai virus-mediated reprogramming (CytoTune2.0) (Thermo Fisher, A16517) under the CUIMC ESCRO guidelines. Every other month, all iPSC lines screened negative for mycoplasma contamination using a MycoAlert PLUS detection kit (Lonza, LT07-710).

### Differentiation of hPSCs into lung epithelial and mesenchymal cells

The directed differentiation protocols were modified from previous protocols to maximize lung mesenchymal cell generation concomitantly with NKX2-1$^+$ lung epithelium (*Hawkins et al., 2017*; *Huang et al., 2014*). Briefly, DE and LPM precursors were induced once seeded hPSC-formed colonies by the Activin induction using the STEMdiff Definitive Endoderm Kit (StemCell Technologies, 05110) for 72 hr. Differentiated cells were dissociated and passaged in Laminin511-coated tissue culture plates in a complete serum-free differentiation medium (cSFDM) (*Chen et al., 2017*). To induce DE and LPM into the anterior foregut endoderm and mesoderm, the cSFDM was supplemented with 10 µM SB431542 (MedChem Express, HY-10431) and 2 µM Dorsomorphin (Tocris, 3093) for 48 hr and 10 µM SB431542 and 2 µM IWP2 (Tocris, 3533) for 24 hr. Cells were then cultured for 7–10 additional days in cSFDM containing 3 µM CHIR99021, 10 ng/ml recombinant human FGF10 (R&D Systems, 345-FG), 10 ng/ml recombinant human KGF (R&D Systems, 251-KG), 10 ng/ml recombinant human BMP4 (R&D Systems, 314-BP), and 50 nM retinoid acid (Sigma-Aldrich, R2625) to induce NKX2-1 positive lung epithelial cells and WNT2$^+$TBX4$^+$ lung mesenchymal cells.

### Immunofluorescence

Before the immunostaining, antigen retrieval was performed using Unmasking Solution (Vector Laboratories, H-3300) for 10 min at around 100°C by microwave. 4–7 µm tissue sections were incubated with primary antibodies in the staining buffer containing 0.025% Triton X-100, and 1% bovine serum

albumin (BSA) overnight at 4°C. Mouse primary antibody staining was done using M.O.M kit (Vector Labs, BMK-2202). Then washed in PBS and incubated with secondary antibodies conjugated with Alexa488, 567, or 647 (Thermo Scientific, 1:400) with NucBlue Fixed Cell Ready Probes Reagent (Hoechst) (Thermo Scientific, R37605) for 1.5 hr, and mounted with ProLong Gold antifade reagent (Invitrogen, P36930). The images were captured by a Zeiss confocal 710 microscope or Leica Stellaris 8 confocal microscopy or DMi8 Leica widefield microscope. The antibodies are listed in *Supplementary file 1*.

## Immunocytochemistry
Cells on culture dishes were fixed with 4% paraformaldehyde (PFA) for 30 min at room temperature (RT), permeabilized, and blocked with staining buffer for 1 hr at RT. Primary antibodies were incubated overnight at 4°C in the staining buffer. After three washes in PBS, secondary antibodies and NucBlue Fixed Cell Ready Probes Reagent (Hoechst) were incubated for 1 hr. The samples were imaged using DMi8 Leica widefield microscope. The antibodies are listed in *Supplementary file 1*.

## RNAScope in situ hybridization
RNA at E18.5 lung sections were stained by RNAScope probes: Mm-Foxa2-T8 (Advanced Cell Diagnostics, #409111-T8) or Negative control (NC) (Advanced Cell Diagnostics, #324341) using the RNAScope HiPlex12 Reagent kit v2 (Advanced Cell Diagnostics, #324419) according to manufacture-provided protocol. Subsequently, sections were incubated with tdTomato antibody for 2 hr at RT, washed in PBS, incubated with secondary antibody conjugated with Alexa488 with NucBlue Fixed Cell Ready Probes Reagent (Hoechst) for 1.5 hr, and mounted with ProLong Gold antifade reagent. The images of *Foxa2* and NC in situ hybridization were captured with the same setting by a Zeiss confocal 710 microscope.

## FCM analyses of mouse lung tissue
Lungs from lineage-tracing mice at E14.5, E18.5, P0, and 4 weeks or lungs from CBC chimeric mice at E14.5, E17.5, and 4 weeks were harvested and prepared for the FCM, as previously described (*Mori et al., 2019*). Briefly, tissues were minced with microscissors, and incubated in 1 ml of pre-warmed dissociation buffer (1 mg/ml DNase [Sigma, DN25], 5 mg/ml collagen [Roche, 10103578001], and 15 U/ml Dispase II [Stemcell Technologies, 7913]) in Hanks' Balanced Salt Solution (HBSS) at 37°C on the rocker with 80 r.p.m. speed, and then neutralized by FACS buffer containing 2% FBS, Glutamax, 2 mM Ethylenediaminetetraacetic acid (EDTA) and 10 mM 4-(2-hydroxyethyl)-1-piperazineethanesulf onic acid (HEPES) in HBSS after the 30–60 min incubation. After filtrating the cells with a 40 µm filter (FALCON, 352235), cell pellets were resuspended with 1 ml of cold RBC lysis buffer (BioLegend, 420301) to lyse the remaining erythrocytes for 5 min on ice, and neutralized by 1 ml cold FACS buffer. After that, it was centrifuged at 350 rcf, 4°C, for 3 min to remove the lysed blood cells. For FCM analysis, one million cells were transferred in 100 µl of FACS buffer supplemented with 0.5 µM Y27632 and then added 2 µl Fc Block (BD Pharmingen, 553141) per sample followed by 10 min incubation on ice. Cells were incubated with the following antibodies: CD31-APC (BioLegend, 102510, 1/50), Epcam-BV711 (BioLegend, 118233, 1/50), or Epcam-BV421 (BioLegend, 118225, 1/50), Aqua Zombie (BioLegend, 423101, 1/100), CD45-BV605 (BioLegend, 103104, 1/50) for 30 min on ice. After staining, cells were washed twice with FACS buffer before resuspending in 500 µl FACS buffer for the subsequent analyses using SONY MA900 or NovoCyte. Compensation was manually performed to minimize the tdTomato signal leakage to the GFP and BV605 channel using FlowJo (ver. 10. 7. 1). Two samples from E14.5 lineage-tracing mice were removed from analysis based on inaccurate staining of live staining. Total four lungs were calculated from six embryos.

## Real-time quantitative RT-PCR
Total RNA was extracted using a Direct-zol RNA MiniPrep Plus kit (Zymo Research, R2072), and cDNA was synthesized using Primescript RT Master Mix (Takara, RR036B). The cDNAs were then used as templates for quantitative RT-PCR analysis with gene-specific primers. Reactions (10 µl) were performed Luna Universal qPCR Master Mix (New England Biolabs, M3003X). mRNA abundance for each gene was determined relative to GAPDH mRNA using the $2^{-\Delta\Delta Ct}$ method. The primers are listed in *Supplementary file 2*. Data were represented as mean ± standard deviation of measurements.

The number of animals or cells per group is provided in the legends. The undetected values in each biological experiment in *Figure 3D* were removed from the graphs.

## EdU assay

100 μg of EdU (Click iT EDU cell proliferation kit, C10337) was injected to pregnant females by intra-peritoneal injection 4 hr prior to analysis. Lungs from lineage-tracing mice at E14.5 and E18.5 were harvested and prepared for the FCM. Then cells were fixed by 4% PFA in PBS and stained by the anufacture's protocol.

## nGFP+ iPSC establishment

E14.5 lung tissues of *Rosa26$^{nTnG/nTnG}$* mice were harvested in a dissociation buffer described above. The dissociated cells were seeded on a 10-cm dish, and only lung fibroblast survived after 1 week in Mouse Embryonic Fibroblast (MEF) medium (*Mori et al., 2019*). The fibroblasts were passaged using Accutase (Innovative Cell Technologies, AT104) and seeded on gelatin (Millipore-Sigma, ES006B)-coated 6-well plates with a density of 0.1 million cells per well. Upon cell attachment, Yamanaka reprogramming factors were induced to iPSCs via Sendai virus using CytoTune2.0 (Thermo Fisher, A16517). To establish nGFP+ iPSCs, the Cre plasmid was transfected using Fugene HD transfection reagent (Promega, E2311), then GFP+tdTomato− live cells were sorted out by SONY MA900, and single clones were expanded.

## Culture of mouse PSC and preparation for CBC donor

For the lung complementation analysis, we injected following PSCs into blastocysts: nGFP+iPSC (established in this study), PSC$^{CAG-GFP}$ (C57BL/6Nx129S6 background, MTI-GlobalStem: cat. no. GSC-5003). Those PSCs were cultured in a2i/VPA/LIF medium on a feeder, as previously reported (*Mori et al., 2019*). These PSCs were passaged and seeded in $10^5$ cells in 6-well plate every 2–3 days. For the CBC donor cell preparation, PSCs were trypsinized 2 min and resuspended in 4 ml cold DMEM + 10% FBS immediately and filtering the cells with a 40 μm filter. Cells were centrifuged at 350 rcf, 4°C, for 3 min, and the supernatant was removed. After being washed with flow buffer containing 0.2% BSA, 1% Glutamax, and 1 μM Y27632, 1 million cells were resuspended in 100 μl flow buffer. The following antibodies were added: Epcam-BV421 (1:50), SSEA1-PE (1:20), CD31-APC (1:20), and Zombie Aqua Fixable Viability Kit (1:100). Epcam$^{high}$SSEA1$^{high}$CD31$^{high}$ cells were sorted by SONY MA900 and subsequently prepared for the injection.

## Blastocyst preparation and embryo transfer

Blastocysts were prepared by mating *Foxa2$^{Cre/Cre}$; Fgfr2$^{flox/+}$, Foxa2$^{Cre/+}$; Fgfr2$^{flox/+}$*, or *Shh$^{Cre/+}$; Fgfr2$^{flox/+}$* males (all 129 × B6 × CD-1 background) with superovulated Fgfr2$^{flox/flox}$; Rosa26$^{tdTomato/tdTomato}$ females (129 × B6 × CD-1 background). Blastocysts were harvested at E3.5 after superovulation (*Mori et al., 2019*). Twenty sorted PSCs were injected into each blastocyst. After the PSC injection, blastocysts were cultured in an M2 medium (Cosmobio) for a few hours in a 37°C, 5% $CO_2$ incubator for recovery. Then, blastocysts were transferred to the uterus of the pseudopregnant foster mother (see *Supplementary file 3*).

## Respiratory function analysis

Respiratory function was measured in 4 weeks chimeric mutant mice (*Foxa2$^{Cre/+}$; Fgfr2* cKO; *Rosa$^{tdTomato}$* or *Foxa2$^{Cre/+}$; Fgfr2* hetero; *Rosa$^{tdTomato}$* injected with PSC$^{CAG-GFP}$) using Buxco FinePointe NAM 2-SiteStaion (DSI) with non-invasive technique (*Pennock et al., 1979*). Values for all measurements represent an average of 5 min monitoring with no rejection index.

## Morphometric analysis

To determine the relative number of specific cell populations, 5 non-overlapping random fields per mouse were analyzed (×40 magnification) after capturing the images by confocal microscopes (see above). We counted the number of GFP+ cells or tdTomato+ cells co-immunostained with specific antibodies for each field. Hoechst co-staining was used to determine the cell number for each lineage as identified by differentiation markers. Those analyses were processed using ImageJ (NIH).

## Statistical analysis

Data analysis was performed using Prism 8. Data acquired by performing biological replicas of two or three independent experiments are presented as the mean ± SD. Statistical significance was determined using a two-tailed *t*-test and unpaired one- or two-way analysis of variance with the Tukey post hoc test. *$p < 0.05$, **$p < 0.01$, ***$p < 0.001$, ****$p < 0.0001$, ns: non-significant.

## Acknowledgements

We thank Zurab Ninish for his technical assistance. We sincerely appreciate the generous support from Dr. Hiromitsu Nakauchi at Stanford University and the considerate support and scientific input from Dr. Wellington Cardoso at the Columbia Center for Human Development (CCHD) and the members of Cardoso's lab and CCHD. We acknowledge the support from the CCHD Medicine Microscopy core (MMC) (NIH S10 OD032447-01), Columbia Stem Cell Initiative (CSCI) Flow Cytometry core, and Genetically Modified Mouse Model Shared Resource (GMMMSR) (P30CA013696) for blastocyst injection. We thank Dr. Heiko Lickert of the Technical University of Munich (TUM) for sharing Foxa2-Cre mice. This work was funded by NIH-NHLBI 1R01 HL148223-01, DoD PR190557, PR191133 to MM, JSPS202080340, and The Uehara Memorial Foundation to AM.

## Additional information

### Funding

| Funder | Grant reference number | Author |
| --- | --- | --- |
| NHLBI Division of Intramural Research | 1R01 HL148223-01 | Munemasa Mori |
| U.S. Department of Defense | PR190557 | Munemasa Mori |
| U.S. Department of Defense | PR191133 | Munemasa Mori |
| Japan Society for the Promotion of Science | JSPS202080340 | Akihiro Miura |
| The Uehara Memorial Foundation | | Akihiro Miura |

The funders had no role in study design, data collection, and interpretation, or the decision to submit the work for publication.

### Author contributions

Akihiro Miura, Formal analysis, Validation, Investigation, Visualization, Methodology, Writing – original draft; Hemanta Sarmah, Junichi Tanaka, Youngmin Hwang, Anri Sawada, Yuko Shimamura, Takehiro Otoshi, Yuri Kondo, Dai Shimizu, Zurab Ninish, Jake Le Suer, Shinichi Toyooka, Jianwen Que, Chyuan-Sheng Lin, Investigation; Yinshan Fang, Resources, Investigation; Nicole C Dubois, Jennifer Davis, Jun Wu, Finn J Hawkins, Resources; Munemasa Mori, Conceptualization, Data curation, Supervision, Funding acquisition, Validation, Investigation, Methodology, Project administration, Writing - review and editing

### Author ORCIDs

Akihiro Miura ⓘ http://orcid.org/0000-0002-1217-7220
Hemanta Sarmah ⓘ http://orcid.org/0000-0002-9374-9269
Youngmin Hwang ⓘ http://orcid.org/0000-0001-5190-6062
Jun Wu ⓘ http://orcid.org/0000-0001-9863-1668
Munemasa Mori ⓘ http://orcid.org/0000-0002-7283-7198

### Ethics

All works using human iPSC lines are derived from deidentified materials here were conducted under the approval of the ethical committee meeting at Columbia University Medical Canter without the requirement of IRB.

All animal experiments were approved by Columbia University Institutional Animal Care and Use Committee protocols (AABF8554) in accordance with US National Institutes of Health guidelines.

### Decision letter and Author response

Decision letter https://doi.org/10.7554/eLife.86105.sa1
Author response https://doi.org/10.7554/eLife.86105.sa2

## Additional files

### Supplementary files

• MDAR checklist

• Supplementary file 1. The antibody list used for the immunostainings and flowcytometry analyses in this article.

• Supplementary file 2. The primer list used for qPCR analysis in the article.

• Supplementary file 3. Summary of PSC donor lines, culture conditions, host mouse strains, and data for chimera formation, lung complementation, and survival in mice subjected to CBC.

### Data availability

scRNA-seq data for mouse gastrulation and early organogenesis described in the manuscript have been analyzed from the deposited database at https://marionilab.cruk.cam.ac.uk/MouseGastrulation2018/. The authors declare that all data supporting the results of this study are available within the paper and the supplementary files.

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
