## [Editor Report]

This study clearly shows the ability to generate whole lungs using a blastocyst complementation assay. The work elegantly uses a Foxa2 deficient by ground to promote generation of lungs from a Foxa2 replete donor. These findings will spur further interest in xenotransplantation approaches for human organs.

---

## [Decision Letter]

**Decision letter after peer review:**

Thank you for submitting your article "Conditional blastocyst complementation of a defective Foxa2 lineage efficiently promotes generation of the whole lung" for consideration by *eLife*. Your article has been reviewed by 2 peer reviewers. The reviewers have opted to remain anonymous.

Essential revisions:

1) Please provide additional data to better reveal the contribution of Foxa2 lineage-labeled Pdgfra+ cells to early mesoderm.

2) Please provide additional data to illustrate the heterogeneity of the cell types in the reported culture systems.

3) Provide quantitation of lineage tracing experiments as outlined by reviewer #2.

4) Provide an additional explanation of the uniqueness of this lung agenesis model as compared to previously reported models i.e. *Fgfr2* null.

*Reviewer #1 (Recommendations for the authors):*

1) The authors use the term bona fide lung generative lineage (BFL) throughout the manuscript. However, the data do not support the concept as such, it should be either be removed or rephrased.

2) The authors state (line 147-148) that the "Foxa2 lineage-labeled Pdgfra+ cells ingress from the primitive streak to the nascent mesoderm regions", providing the data shown in Figure 1D as evidence. While it seems reasonable to conclude that this is the case, a single static confocal image isn't sufficient data upon which to draw this conclusion.

3) In Figure 3B, the authors highlight "distal lung bud epithelium". While the Nkx2.1+ cells do also appear to express *SOX9*, the majority of *SOX9*+ cells in the culture do not appear to express *SOX9*. This should be clarified. Are most of the *SOX9*-expressing cells in this culture system in fact mesenchyme, which only small clusters of Nkx2.1+/*Sox9*+ lung epithelium?

4) The authors should more explicitly outline how blastocyst complementation using the current knockout mouse model (Foxa2-Cre; *Fgfr2* fl/fl) significantly adds to the existing body of literature where blastocyst complementation has already been performed in mice with a severe lung hypoplasia or agenesis phenotype (Nkx2.1-/-, Fgf10-/-, conditional epithelial-specific ablation of Ctnnb1 or *Fgfr2*).

5) In Figures 4D and Tables 1-2 blastocyst complementation following conditional loss of *Fgfr2* with either the Foxa2-Cre or Shh-Cre are compared, but no statistics are provided. Specifically, the authors claim that in contrast to the Shh-Cre animals, in the Foxa2-Cre mice that "almost the entire lung epithelial, mesenchymal and endothelial cells populations at E17.5 was composed exclusively of nGFP+iPSCs". However, the numbers provided don't seem to support this claim. For example, just looking at the mesenchyme Foxa2-Cre (92.6, 90, 89.8, 87.3, 57.6) versus Shh-Cre (71.7, 76.5). While it is true that the mean percentage is slightly higher in the Foxa2 mice, although the mean is in fact slightly higher (83.5% versus 74.1%), this is both not statistically significant and even if it was, a relatively small difference given that it is a major claim being made to support the significance of the findings. These data need to be expanded and clarified, particularly in relation to the previous comment (#4).

6) It is interesting that despite the lung agenesis phenotype, the other organs appear to be relatively spared (Supplemental Figure 2). It would be interesting to explore this further.

*Reviewer #2 (Recommendations for the authors):*

The conclusions of this paper are mostly well supported by data, but some aspects of data analysis need to be clarified and extended.

1). Figure 4D showed that ~20% of mesenchyme and endothelium cells are still derived from host cells, it will be interesting for authors to investigate whether host cells contribute to certain types of mesenchyme and endothelium cells or random contributed.

2). For figure 1B, the text describes Pdgfrb+ pulmonary mesenchyme, however, the figure show Pdgfra staining.

3). The author should discuss the reason that they did not observe agenesis phenotype in other major internal organs in Foxa2-driven *Fgfr2*-deficient mice.

---

## [Author Response]

Essential revisions:1) Please provide additional data to better reveal the contribution of Foxa2 lineage-labeled Pdgfra+ cells to early mesoderm.

We performed additional IF (immunofluorescent)-confocal morphometric analyses at the primitive streak (PS) stage to better reveal the contribution of Foxa2 lineage-labeled Pdgfra+ cells to early mesoderm. We found that 80% of gastrulating PS cells expressed Pdgfra protein expression, of which about 20% were co-labeled with tdTomato+ Foxa2 lineage. This proportion is nearly the same proportion of E14.5 Foxa2-lineage labeled cells in lung mesenchyme. This indicates that lung mesenchyme precursor marked by Foxa2-linage+ Pdgfra+ cells originates mid-streak during gastrulation before the lateral plate mesoderm (LPM) formation. These data were added in the new Figure 1F, and 1G and the manuscript lines 156-162.

2) Please provide additional data to illustrate the heterogeneity of the cell types in the reported culture systems.

We additionally performed IF morphometric analyses across the time point to reveal better the cell heterogeneity of lung progenitors and precursors in the human lung-directed differentiation protocol. To reflect this finding of the heterogeneity of lung progenitors based on the morphometric analyses, we replaced Figure 3C with the new Figure 3 —figure supplement 1 and described it in the manuscript, lines 221-230.

It showed that NKX2-1+ cells preceded to appear on the bud-like structure on day 6 and followed by the *Sox9* expression. On day 15, most budding cells turned on NKX2-1 and *SOX9*. Further morphometric analyses also found that non-budding lung progenitors gradually turned on TBX4, and about 50% of the non-budding cells expressed TBX4, which made the heterogeneity of lung progenitors. Non-TBX4 positive cells are possibly lung progenitors that do not express TBX4 yet or another cell type. This finding was added in new Figure 3 —figure supplement 1 and the manuscript, lines 221-230.

3) Provide quantitation of lineage tracing experiments as outlined by reviewer #2.

We quantified the lineage-tracing results by morphometric analyses described in Figures 1C and 1G. We provided the quantification of Foxa2 lineage tracing studies in early embryogenesis and removed the unqualified results from Figure 1 (Figure 1F and 1G and Figure 1C), and the manuscript was corrected in lines 136-144 and 154-162.

Regarding Figure 1C, we have tried to have more numbers of embryos for these analyses using Pdgfra^CreERT2^; Rosa ^tdTomato/+^ mice. However, we often encountered embryo miscarriage due to the effect of Tamoxifen, even with the titration of tamoxifen or using the co-injection of progesterone (Nikita et al., 2019). Through more than twenty times experimental trials of Tm injection, we finally obtained a total of four embryos, three at E12.5 and one at E14.5.

E14.5 lung immunostaining analysis showed that the entire lung mesenchyme was labeled with tdTomato but not much in the epithelium (Figure 1A). Two E12.5 lungs showed entire labeling of tdTomato, but one embryo showed around 50% labeling in the mesenchyme and no labeling in the epithelium. We thought that this was because of the short time window of Pdgfra expression during gastrulation. This result corresponds to directed differentiation using human iPSC and the deposited scRNAseq analyses described in Figure 5—figure supplement1A; Pdgfra temporally expressed during gastrulation but not spontaneously expressed in the later stage. This data was outlined in the manuscript, lines 136-144.

4) Provide an additional explanation of the uniqueness of this lung agenesis model as compared to previously reported models i.e. Fgfr2 null.

With this constructive suggestion from the reviewers, we extensively analyzed the lungs and other organs. This approach led us to discover the novel insights of this model (see the following points), lacking in the Shh-based CBC model.

1. We also increased the number of N of the biological replicates for the flow cytometry analyses in comparing the Shh and Foxa2-based CBC models. Although there is a clear tendency of the increased complementation proportion in the Foxa2 model than Shh models as described in the new Figure 4D-4E and Figure 4—figure supplement 3C-3D, we could not have the statistical significance. Thus, we quantified the chimerism with mesenchyme markers by morphometric analyses in the chimeric lungs of the Shh vs. Foxa2 -based CBC model. It revealed that the chimerism in Sma+ airway smooth muscle cells, CD31+ capillary, and NG2+ pericyte showed significantly higher chimerism in the Foxa2 model compared to the Shh model. This data was included in the new Figure 4F-H and Figure 4—figure supplement 4 – outlined in the manuscript, lines 295-300.

2. We found that the knockout model analysis of Foxa2^Cre/+^; Fgfr2^cnull^ showed shorter trachea compared to Shh^Cre/+^; Fgfr2^cnull^ trachea. Foxa2^Cre/+^; Fgfr2^cnull^ showed truncated trachea in the middle of thoracic cavity all the time and all the time stopped elongation above the aortic arch, meantime Shh^Cre/+^; Fgfr2^cnull^ showed entire trachea and truncated in the middle of the bronchus. This severe malformation phenocopies the Fgf10 knockout mice. This could be due to the distinct lineage labeling of Shh and Foxa2 during primordial lung formation. Further investigation is required in future studies. When we analyzed the chimerism in the tracheal epithelium of Foxa2^Cre/+^; Fgfr2^cnull^ injected with donor cells, significantly higher chimerism in the tracheal epithelium but not in the tracheal mesenchyme, this result is included in Figure 4—figure supplement 5 and outlined in the manuscript, lines 300-304.

3. We accidentally found that Foxa2^Cre/+^; Fgfr2^cnull^ showed thymus agenesis phenotype as described in the new Figure 4 —figure supplement 2, but not in Shh^Cre/+^; Fgfr2^cnull^. *Fgfr2* is known to be necessary for thymus formation (Dooley et al., 2007). When we analyzed chimerism in the thymic epithelium, Foxa2^Cre/+^; Fgfr2^cnull^ injected with donor cells showed significantly higher chimerism. These new findings and discussion were included in Figure 4 —figure supplement 2, and the significance was described in the manuscript, lines 307-313 and 431-456.

Reviewer #1 (Recommendations for the authors):1) The authors use the term bona fide lung generative lineage (BFL) throughout the manuscript. However, the data do not support the concept as such, it should be either be removed or rephrased.

We deleted the BFL concept and the sentences from the entire manuscript.

2) The authors state (line 147-148) that the "Foxa2 lineage-labeled Pdgfra+ cells ingress from the primitive streak to the nascent mesoderm regions", providing the data shown in Figure 1D as evidence. While it seems reasonable to conclude that this is the case, a single static confocal image isn't sufficient data upon which to draw this conclusion.

We deleted the description, and we showed the additional evidence with the quantification results in Figure 1E-1G outlined in the manuscript of lines 154-162.

3) In Figure 3B, the authors highlight "distal lung bud epithelium". While the Nkx2.1+ cells do also appear to express SOX9, the majority of SOX9+ cells in the culture do not appear to express SOX9. This should be clarified. Are most of the SOX9-expressing cells in this culture system in fact mesenchyme, which only small clusters of Nkx2.1+/Sox9+ lung epithelium?

The previous image did not explain well the directed differentiation heterogeneity. Therefore, we performed time course IF analyses and their quantification by morphometric analyses for the directed differentiation. It was described in Figure 3 —figure supplement 1 and outlined in the manuscript of lines 215-229. In the middle of differentiation, bud structure appeared in the culture dish. Those NKX2-1 and *SOX9* expression patterns were initially random around days 6~8 and mosaic but later homogenously expressed both NKX2-1 and *SOX9* around days 14~15. To reflect this finding of the heterogeneity of lung progenitors based on the morphometric analyses, we replaced Figure 3B.

4) The authors should more explicitly outline how blastocyst complementation using the current knockout mouse model (Foxa2-Cre; Fgfr2 fl/fl) significantly adds to the existing body of literature where blastocyst complementation has already been performed in mice with a severe lung hypoplasia or agenesis phenotype (Nkx2.1-/-, Fgf10-/-, conditional epithelial-specific ablation of Ctnnb1 or Fgfr2).

Our new Foxa2-lineage-based CBC model mice showed novel evidence of the co-generation of lung and thymus. We also added evidence that those rescued mice of the Foxa2-lineage-based CBC model survived until adulthood with normal lung function. Furthermore, our overall analyses showed relatively high efficacy of functional lung complementation in adults than the previous reports (see Supplementary File 3). All of this evidence was not shown in the previous complementation method.

5) In Figures 4D and Tables 1-2 blastocyst complementation following conditional loss of Fgfr2 with either the Foxa2-Cre or Shh-Cre are compared, but no statistics are provided. Specifically, the authors claim that in contrast to the Shh-Cre animals, in the Foxa2-Cre mice that "almost the entire lung epithelial, mesenchymal and endothelial cells populations at E17.5 was composed exclusively of nGFP+iPSCs". However, the numbers provided don't seem to support this claim. For example, just looking at the mesenchyme Foxa2-Cre (92.6, 90, 89.8, 87.3, 57.6) versus Shh-Cre (71.7, 76.5). While it is true that the mean percentage is slightly higher in the Foxa2 mice, although the mean is in fact slightly higher (83.5% versus 74.1%), this is both not statistically significant and even if it was, a relatively small difference given that it is a major claim being made to support the significance of the findings. These data need to be expanded and clarified, particularly in relation to the previous comment (#4).

We extended the analysis of the complemented lungs with an increased number of N (biological replicates). In flow cytometry data analyses, we still did not observe the statistical significance yet, while we observed the tendency of the higher proportion of complementation in the Foxa2-lineage-based CBC model. Therefore, we quantified the lung mesenchyme complementation by the IF morphometric analyses. It revealed that the chimerism in Sma+ airway smooth muscle cells, CD31+ capillary, and NG2+ pericyte showed significantly higher chimerism in the Foxa2 model compared to the Shh model. We included this data in Figures 4F-H and Figures 4—figure supplement 4, and the manuscript of lines 288-293. As described above, we also added novel evidence of the co-generation of the lung and thymus, which was not defined in the previous papers.

6) It is interesting that despite the lung agenesis phenotype, the other organs appear to be relatively spared (Supplemental Figure 2). It would be interesting to explore this further.

As described above, we included the analysis of Foxa2^Cre/+^; Fgfr2^cnull^ phenotype in new Figures 4 —figure supplement 1C and 2A outlined in the manuscript, lines 266-276.

Reviewer #2 (Recommendations for the authors):The conclusions of this paper are mostly well supported by data, but some aspects of data analysis need to be clarified and extended.1). Figure 4D showed that ~20% of mesenchyme and endothelium cells are still derived from host cells, it will be interesting for authors to investigate whether host cells contribute to certain types of mesenchyme and endothelium cells or random contributed.

We further analyzed the chimerism based on cell types and found that Sma+ airway smooth muscle, Pecam+ capillary, and NG2+ pericyte showed higher contributions from injected donor cells. The results were included in Figures 4F-H and Figures 4—figure supplement 4 and in the manuscript lines 295-300.

2). For figure 1B, the text describes Pdgfrb+ pulmonary mesenchyme, however, the figure show Pdgfra staining.

We apologize for the careless mistake. We corrected the term described in the manuscript line 140.

3). The author should discuss the reason that they did not observe agenesis phenotype in other major internal organs in Foxa2-driven Fgfr2-deficient mice.

This phenotype was based on the combination of Cre driver and *Fgfr2* necessity for organ development. We included this discussion in the manuscript lines 399-404.